# Divergent regulation of KCNQ1/E1 by targeted recruitment of protein kinase A to distinct sites on the channel complex

**Xinle Zou**[1†], **Sri Karthika Shanmugam**[2†], **Scott A Kanner**[3†], **Kevin J Sampson**[1], **Robert S Kass**[1], **Henry M Colecraft**[1,3]*

[1]Department of Molecular Pharmacology and Therapeutics, Columbia University, New York, United States; [2]Department of Physiology and Cellular Biophysics, Columbia University, New York, United States; [3]Doctoral Program in Neurobiology and Behavior, Columbia University, New York, United States

*For correspondence:
hc2405@cumc.columbia.edu

†These authors contributed equally to this work

**Abstract** The slow delayed rectifier potassium current, $I_{Ks}$, conducted through pore-forming Q1 and auxiliary E1 ion channel complexes is important for human cardiac action potential repolarization. During exercise or fright, $I_{Ks}$ is up-regulated by protein kinase A (PKA)-mediated Q1 phosphorylation to maintain heart rhythm and optimum cardiac performance. Sympathetic up-regulation of $I_{Ks}$ requires recruitment of PKA holoenzyme (two regulatory – RI or RII – and two catalytic Cα subunits) to Q1 C-terminus by an A kinase anchoring protein (AKAP9). Mutations in Q1 or AKAP9 that abolish their functional interaction result in long QT syndrome type 1 and 11, respectively, which increases the risk of sudden cardiac death during exercise. Here, we investigated the utility of a targeted protein phosphorylation (TPP) approach to reconstitute PKA regulation of $I_{Ks}$ in the absence of AKAP9. Targeted recruitment of endogenous Cα to E1-YFP using a GFP/YFP nanobody (nano) fused to RIIα enabled acute cAMP-mediated enhancement of $I_{Ks}$, reconstituting physiological regulation of the channel complex. By contrast, nano-mediated tethering of RIIα or Cα to Q1-YFP constitutively inhibited $I_{Ks}$ by retaining the channel intracellularly in the endoplasmic reticulum and Golgi. Proteomic analysis revealed that distinct phosphorylation sites are modified by Cα targeted to Q1-YFP compared to free Cα. Thus, functional outcomes of synthetically recruited PKA on $I_{Ks}$ regulation is critically dependent on the site of recruitment within the channel complex. The results reveal insights into divergent regulation of $I_{Ks}$ by phosphorylation across different spatial and time scales, and suggest a TPP approach to develop new drugs to prevent exercise-induced sudden cardiac death.

## Editor's evaluation

This important study finds that the location of recruitment of a protein kinase to an ion channel can change the complement of residues modified, leading to channel function being altered by a qualitatively distinct mechanism. Evidence for this major claim is compelling.

## Introduction

Regulating protein functional expression by targeted induced proximity of enzymes (TIPE) is a burgeoning field with great promise for developing therapeutics to conventionally undruggable targets. The leading edge of this broad concept has been targeted protein degradation (TPD) with small-molecule proteolysis-targeting chimeras (PROTACs) which work by recruiting endogenous E3 ubiquitin ligases to chosen proteins (*Sakamoto et al., 2001*; *Schneekloth et al., 2004*; *Nalawansha*

*and Crews, 2020*). Recent promising phase I and II clinical trials of targeted proteins degraders represent a realization of the potential of the overall TIPE approach (*Békés et al., 2022*). Beyond TPD, there have been nascent efforts to expand the TIPE concept to other physiologically important enzyme classes, including targeted protein stabilization (TPS) via recruitment of deubiquitinases (*Kanner et al., 2020*; *Henning et al., 2022*), targeted dephosphorylation using induced proximity of a phosphatase (PhosTACs) (*Chen et al., 2021*), lysosome-targeting chimeras that enable degradation of extracellular proteins (*Banik et al., 2020*), and targeted protein acetylation (*Wang et al., 2021*).

Posttranslational regulation of proteins by phosphorylation is a prominent mechanism for regulating cell biology and physiology that is timely for adaptation to the TIPE idea and has been broached with the development of PhosTACs (*Chen et al., 2021*) and phosphorylation inducing chimeric small molecules (PHICS) (*Siriwardena et al., 2020*). The human genome contains ~500 kinases most of which phosphorylate hydroxyl groups on either serine/threonine or tyrosine residues in proteins (*Fabbro et al., 2015*). Site-specific phosphorylation of proteins can lead to diverse outcomes including multidimensional regulation of functional activity (*Cohen, 2000*), protein-protein interactions (*Liu et al., 2020*; *Pennington et al., 2018*), protein stability (*Xu et al., 2009*), and subcellular localization (*Liu et al., 2020*; *El Amri et al., 2018*; *Yang et al., 2007*). Moreover, individual proteins typically have multiple residues that may be phosphorylated raising the possibility of actuating distinctive functional outcomes depending on the particular complement of serine, threonine, or tyrosine residues modified under different conditions (*Cohen, 2000*). Induced recruitment of a kinase to a target may potentially be used to either reconstitute physiologically cognate responses or to realize de novo synthetic regulation of protein activity. Overall, by comparison to TPD or stabilization, regulating proteins by targeted protein phosphorylation (TPP) is a potentially more complex prospect, requiring deeper insights into the rules required to achieve specific desired outcomes.

Here, we explore these dimensions of TPP by focusing on a cardiac ion channel comprised of pore-forming Q1 and auxiliary E1 subunits. In human heart, Q1/E1 channels give rise to the slowly activating delayed rectifier potassium current, $I_{Ks}$, which is essential for normal cardiac action potential repolarization (*Sanguinetti et al., 1996*). During sympathetic activation of the heart that occurs during exercise or the fight-or-flight response, β-adrenergic up-regulation of $I_{Ks}$ occurs via protein kinase A (PKA)-mediated phosphorylation of Q1 on N-terminus serine residues (Ser27 and Ser92) (*Kurokawa et al., 2003*; *Marx et al., 2002*; *Lundby et al., 2013*). This effect is crucial to counterbalance the increase in L-type calcium current, that also occurs during sympathetic activation of the heart, to maintain the action potential duration (APD) in an appropriate range (*Banyasz et al., 2014*; *Volders et al., 2003*; *Gadsby, 1983*). Loss-of-function mutations in KCNQ1 cause long QT syndrome type 1 (LQT1) due to an increase in the APD that is exacerbated in exercise, thereby elevating the risk of ventricular tachycardias (torsade de pointes) and sudden cardiac death (*Schwartz et al., 2012*). β-Adrenergic regulation of $I_{Ks}$ is critically dependent on the A kinase anchoring protein, AKAP9 (yotiao), which binds to the C-terminus of Q1 and acts as a scaffold to recruit PKA holoenzyme (comprised of two regulatory RI or RII and two catalytic Cα subunits) to the channel complex (*Marx et al., 2002*). Several LQT1 mutations in Q1 C-terminus occur along the binding interface with AKAP9 and may disrupt this crucial protein-protein interaction (*Marx et al., 2002*; *Aromolaran et al., 2014*; *Tester et al., 2005*; *Howard et al., 2007*). Moreover, mutations in AKAP9 that diminish the interaction with Q1 act as genetic modifiers of LQT1 and cause LQT11 (*Schwartz et al., 2012*; *Chen et al., 2007*).

We hypothesized that we could exploit TIPE to reconstitute PKA-mediated regulation of $I_{Ks}$ in the absence of AKAP9. If successful, this could pave the way to development of PHICS as therapeutics for LQTS. We utilized an anti-GFP nanobody (nano) to direct PKA RIIα or Cα subunits to YFP-tagged Q1/E1 channel complexes. Targeted recruitment of endogenous Cα to E1-YFP using nanoRIIα fusion protein enabled acute PKA-mediated enhancement of $I_{Ks}$, reconstituting the physiological regulation of the channel complex. Simply overexpressing free Cα constitutively recapitulated PKA phosphorylation of Q1 and functional regulation of $I_{Ks}$ which were not further enhanced by targeting nanoCα to E1-YFP. In sharp contrast, nano-mediated tethering of either RIIα or Cα to Q1-YFP eliminated $I_{Ks}$ by retaining Q1 intracellularly in the endoplasmic reticulum (ER) and Golgi compartments. Thus, functional outcomes of synthetically recruited PKA on $I_{Ks}$ regulation is critically dependent on the site of recruitment within the Q1/E1 channel complex.

## Results

### Targeting RIIα to E1 reconstitutes acute PKA modulation of $I_{Ks}$, whereas anchoring it to Q1 inhibits basal current

We reconstituted $I_{Ks}$ by co-expressing Q1+E1 in Chinese hamster ovary (CHO) cells. To examine whether the reconstituted currents are modulated by acute PKA activation, we measured the time course of $I_{Ks}$ amplitude after breakthrough to the whole-cell configuration with intracellular solution±cAMP/okadaic acid in the patch pipette. In control cells expressing Q1+E1-YFP+naked nanobody (nano) (*Figure 1A*), exemplar $I_{Ks}$ did not increase when compared between immediately after breakthrough and 3 min after dialysis with patch pipette solution either lacking (control) or containing cAMP/OA (*Figure 1B*). In diary plots of population data, control cells showed a monotonic rundown of $I_{Ks}$ amplitude that was similar between whether the patch solution contained cAMP/OA or not (*Figure 1C*). These results are consistent with previously published work showing that acute PKA regulation of $I_{Ks}$ does not occur in heterologous cells in the absence of co-expressed AKAP9 (yotiao) (*Marx et al., 2002*).

We sought to determine whether we could utilize a nanobody-based targeted recruitment approach to reconstitute acute cAMP-induced PKA regulation of $I_{Ks}$ independent of AKAP9. To enable recruitment of endogenous Cα in a configuration where it is basally inactive and can be acutely activated by cAMP near the channel complex, we fused anti-GFP nanobody to the regulatory subunit (RIIα) of PKA to generate nanoRIIα. In cells expressing Q1+E1-YFP+nanoRIIα (*Figure 1D*), exemplar $I_{Ks}$ did not show an increase in current amplitude after breakthrough to the whole-cell configuration if the pipette solution lacked cAMP/OA (*Figure 1E*, *left*). In sharp contrast, when cAMP/OA was present in the patch pipette, exemplar $I_{Ks}$ displayed a significant increase in current after 3 min of dialysis with intracellular solution (*Figure 1E*, *right*). In population time course data, the temporal evolution of normalized current amplitude displayed a clear-cut divergence between recordings obtained with or without cAMP/OA in the patch pipette solution ($I_{3min}/I_0$=0.9615 ± 0.025, *n*=14 for Q1+E1-YFP+nanoRIIα without cAMP+OA; $I_{3min}/I_0$=1.0812 ± 0.030, *n*=12, for Q1+E1-YFP+nanoRIIα with cAMP+OA; p=0.0054, unpaired t-test) (*Figure 1F*). The magnitude of the observed response in the diary plots is comparable to the normalized enhancement of $I_{Ks}$ current observed with cAMP+OA in cells expressing Q1+E1+AKAP9 (*Kurokawa et al., 2004*). Controls were measured interleaved with the experiment group. In additional control experiments, cells expressing untagged Q1+E1+nanoRIIα displayed no increase or divergence in $I_{Ks}$ amplitude between recordings in the absence or presence of cAMP/OA in the patch pipette solution (*Figure 1—figure supplement 1*). Overall, these data demonstrate successful reconstitution of acute PKA regulation of $I_{Ks}$ by using a nanobody to tether the regulatory RIIα subunit to E1-YFP in the channel complex.

Does the site of PKA recruitment to the Q1/E1 channel complex matter with respect to functional outcomes? To address this question, we attached YFP to Q1 C-terminus instead of E1 and co-expressed the channel with either naked nanobody (control; nano) or nanoRIIα (*Figure 1G*). Control cells expressing Q1-YFP+E1+nano displayed robust basal $I_{Ks}$ (*Figure 1H*). Unexpectedly, co-expressing nanoRIIα with Q1-YFP/E1 yielded a substantially suppressed basal $I_{Ks}$, suggesting that the site of PKA recruitment to the channel complex is critical in determining functional outcomes (*Figure 1H and I*).

### Targeting PKA-Cα to either Q1 or E1 with a nanobody yields divergent functional outcomes on reconstituted $I_{Ks}$

AKAP9 enables acute PKA regulation of $I_{Ks}$ by acting as a scaffold that increases the effective local concentration of PKA holoenzyme in the vicinity of the channel complex (*Marx et al., 2002*; *Langeberg and Scott, 2015*). Accordingly, we wondered whether simply overexpressing free Cα would suffice to reconstitute aspects of PKA regulation of $I_{Ks}$. Indeed, western blotting indicated that co-expressing Cα led to an increase in Q1 phosphorylation (normalized pQ1/Q1 is increased 1.516-fold for cell expressing Cα compared to control) (*Figure 2A and B*). Next, we compared key gating parameters of currents recorded from CHO cells expressing Q1+E1-YFP, either with or without Cα co-expression. Current vs voltage (*I-V*) curves indicated that by comparison to currents obtained with Q1+E1-YFP alone, those recorded from cells co-expressing Cα displayed a hyperpolarizing shift in the voltage dependence of activation ($V_{0.5,act}$ = 34.5 ± 3.6 mV, *n*=13 for Q1+E1-YFP and $V_{0.5,act}$ = 25.0 ± 2.7, *n*=13 for Q1+E1-YFP+Cα, p=0.02, unpaired t-test) (*Figure 2C*), and a trend toward a slower rate of

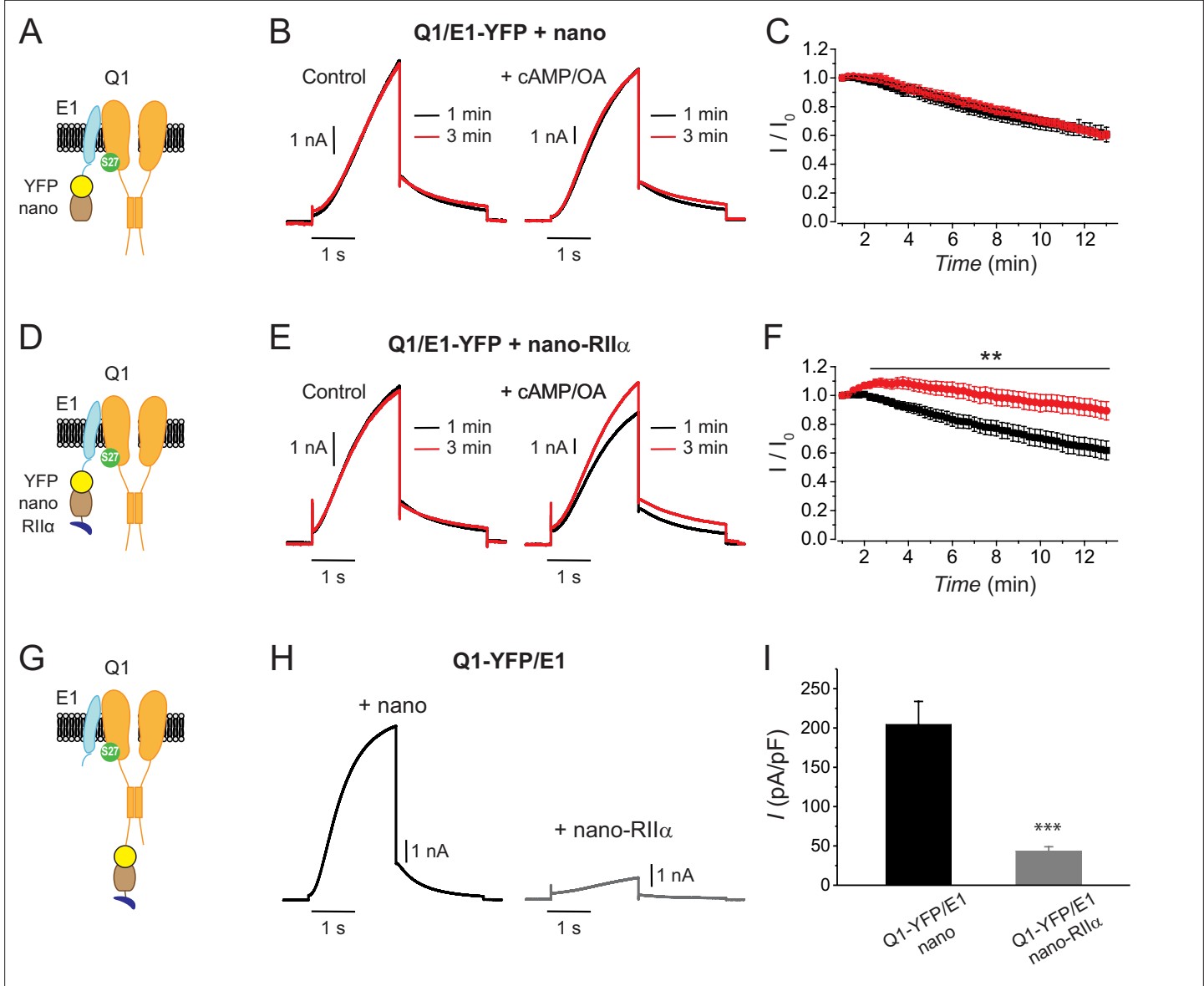

**Figure 1.** Differential functional effects of nanoRIIα targeted to either KCNE1 or KCNQ1 on $I_{Ks}$. (**A**) Cartoon showing targeting GFP/YFP nanobody (nano) to Q1/E1 channel complex via a YFP tag on E1. (**B**) Exemplar $I_{Ks}$ traces elicited by test pulses (+60 mV, –40 mV return) reconstituted in Chinese hamster ovary (CHO) cells expressing Q1/E1-YFP+nano at 1 min (black traces) or 3 min (red traces) after break-in to whole-cell configuration. Cells were dialyzed with internal solution either lacking (*left*) or including (*right*) 0.2 mM cAMP+0.2 µM okadaic acid (cAMP/OA). (**C**) Diary plot of population tail-current amplitudes (mean ± SEM) vs time with cAMP/OA either lacking (black symbols, *n*=10) or included (red symbols, *n*=11) in the patch pipette solution. (**D–F**) Cartoon, exemplar currents and population tail-current amplitude vs time for CHO cells expressing Q1/E1-YFP+nanoRIIα. Same format as (**A–C**). **\*\*p<0.01, two-tailed unpaired t-test. (**G**) Cartoon showing nanoRIIα targeting to Q1/E1 channel complex via YFP tag on Q1. (**H**) Exemplar $I_{Ks}$ traces reconstituted in CHO cells expressing Q1-YFP/E1 with either nano (*left*) or nanoRIIα (*right*). (**I**) Population current densities (nano, *n*=26; nanoRIIα, *n*=17). \*\*\*p<0.001, two-tailed unpaired t-test.

The online version of this article includes the following source data and figure supplement(s) for figure 1:

**Source data 1.** Differential functional effects of nanoRIIα targeted to either KCNE1 or KCNQ1 on $I_{Ks}$.

**Figure supplement 1.** NanoRIIα does not reconstitute protein kinase A (PKA) regulation of $I_{Ks}$ when co-expressed with untagged KCNQ1+KCNE1.

**Figure supplement 1—source data 1.** NanoRIIα does not reconstitute protein kinase A (PKA) regulation of $I_{Ks}$ when co-expressed with untagged KCNQ1+KCNE1.

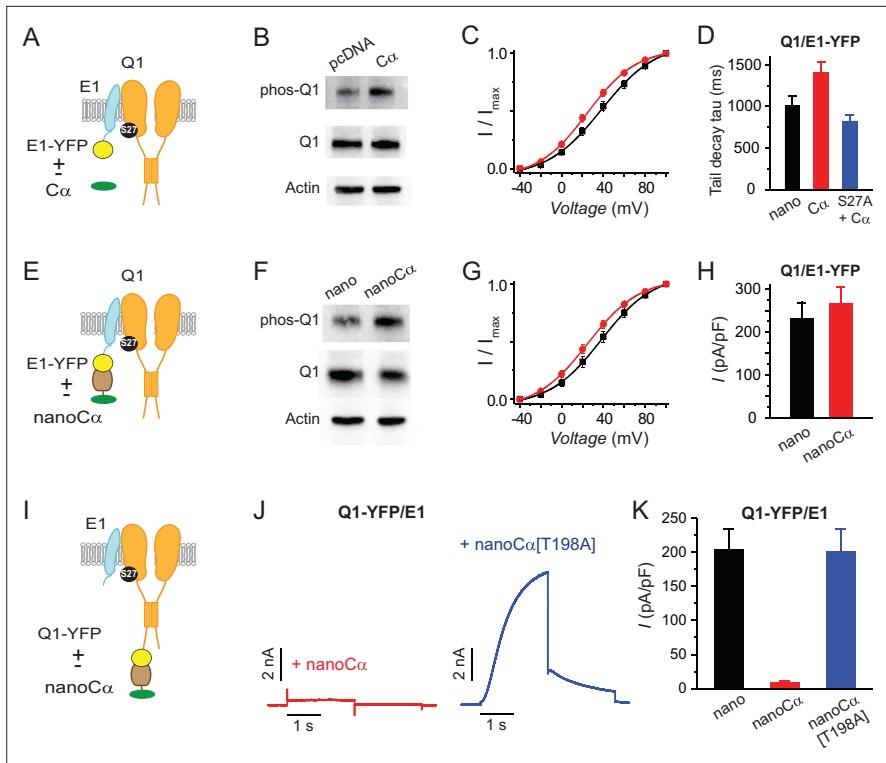

**Figure 2.** Differential functional effects of nano-Cα targeted to either Q1 or E1 on $I_{Ks}$. (**A**) Cartoon showing Q1/E1-YFP complex co-expressed with or without free protein kinase A (PKA) Cα subunit. (**B**) Representative immunoblots of lysates from HEK293 cells co-expressing Q1/E1-YFP with either empty pcDNA3.1 vector or free Cα. Anti-pKCNQ1 (top) detects phosphorylated KCNQ1-S27, anti-KCNQ1 (middle) detects total KCNQ1, and anti-actin (bottom) detects total actin. $N=1$. (**C**) $I_{Ks}$ activation curves in Chinese hamster ovary (CHO) cells co-expressing Q1, E1-YFP with either empty pcDNA3.1 vector (black symbols, $n=13$) or free PKA Cα (red symbols, $n=13$). (**D**) Tail-decay times for currents recorded from cells co-expressing Q1/E-YFP+yotiao and either nano or free PKA Cα, or cells co-expressing Q1[S27A]/E1-YFP+yotiao and free PKA Cα ($p=0.0532$, one-way ANOVA). (**E–H**) Cartoon, immunoblots, $I_{Ks}$ activation curves, and population current densities of Q1/E1-YFP complex expressed with either nano ($n=10$) or nanoCα ($n=10$). (**I**) Cartoon showing targeting of nanoCα to Q1/E1 complex via YFP tag on Q1 C-terminus. (**J**) Exemplar $I_{Ks}$ traces from CHO cells co-expressing Q1-YFP/E1 with either nanoCα (*left*) or catalytically inactive nanoCα [T198A] mutant (*right*). (**K**) Population current densities (nano, $n=26$; nanoCα, $n=19$; nanoCα[T198A], $n=10$).

The online version of this article includes the following source data and figure supplement(s) for figure 2:

**Source data 1.** Differential functional effects of nano-Cα targeted to either Q1 or E1 on $I_{Ks}$.

**Source data 2.** Differential functional effects of nano-Cα targeted to either Q1 or E1 on $I_{Ks}$.

**Figure supplement 1.** Evidence that nanoCα but not free protein kinase A (PKA) Cα is recruited to E1-YFP in the Q1/E1-YFP channel complex.

**Figure supplement 1—source data 1.** Full immunoblots of experiments showing nanoCα but not free PKA Cα is recruited to E1-YFP in the Q1/E1-YFP channel complex.

**Figure supplement 2.** NanoCα targeted to the C-terminus of TASK1 via a GFP tag does not inhibit K[+] current.

**Figure supplement 2—source data 1.** NanoCα targeted to the C-terminus of TASK1 via a GFP tag does not inhibit K[+] current.

---

tail current deactivation (in the presence of yotiao) (*Figure 2D*), two signatures of PKA regulation of $I_{Ks}$ that is mediated via phosphorylation of Ser27 in Q1 (*Marx et al., 2002*; *Chen et al., 2005*). Consistent with this, Q1[S27A] reversed the trend toward Cα-induced decreased rate of tail current deactivation observed with wild-type Q1 (*Figure 2D*).

With the impact of free Cα as a baseline, we next assessed how nanobody-mediated recruitment of Cα to either E1 or Q1 would impact reconstituted $I_{Ks}$. We fused Cα to anti-GFP nanobody to generate nanoCα and co-expressed it with Q1+E1-YFP. In this configuration, Cα is recruited to the tagged

E1 subunit in the channel complex (*Figure 2E*), as confirmed in a pull-down assay (*Figure 2—figure supplement 1*). Compared to control cells expressing Q1+E1-YFP+nano, channels co-expressed with nanoCα displayed an increased phosphorylation of Q1 (normalized pQ1/Q1 is increased 2.2-fold in cells co-expressing nanoCα compared to controls expressing nano) (*Figure 2F*), a leftward shift in the voltage dependence of activation ($V_{0.5,act}$ = 34.1 ± 4.4 mV, $n$=10 for Q1+E1-YFP+nano and $V_{0.5,act}$ = 25.2 ± 3.4, $n$=10 for Q1+E1-YFP+nanoCα, p=0.049, unpaired, t-test) (*Figure 2G*), and a trend toward a small augmentation of basal current amplitude ($I_{avg}$ = 232.1 ± 36.44 mV, $n$=11 for Q1+E1-YFP+nano and $I_{avg}$ = 267.24 ± 37.12, $n$=16 for Q1+E1-YFP+nanoCα, p=0.5054) (*Figure 2H*). By contrast, recruiting Cα to Q1 (Q1-YFP+E1+nanoCα) resulted in a drastic elimination of $I_{Ks}$ (*Figure 2J*). To ascertain that the observed effect is due to the activity of the targeted kinase, we introduced a T198A mutation into Cα that renders it catalytically dead. Co-expressing nanoCα[T198A] with KCNQ1-YFP/KCNE1 yielded robust currents that were similar in amplitude to control (nano), indicating that intact kinase activity is necessary for the inhibitory effect observed with nanoCα (*Figure 2K*). Co-expressing nanoCα with TASK-4-GFP, a two-pore domain K⁺ channel, did not decrease whole-cell current density compared

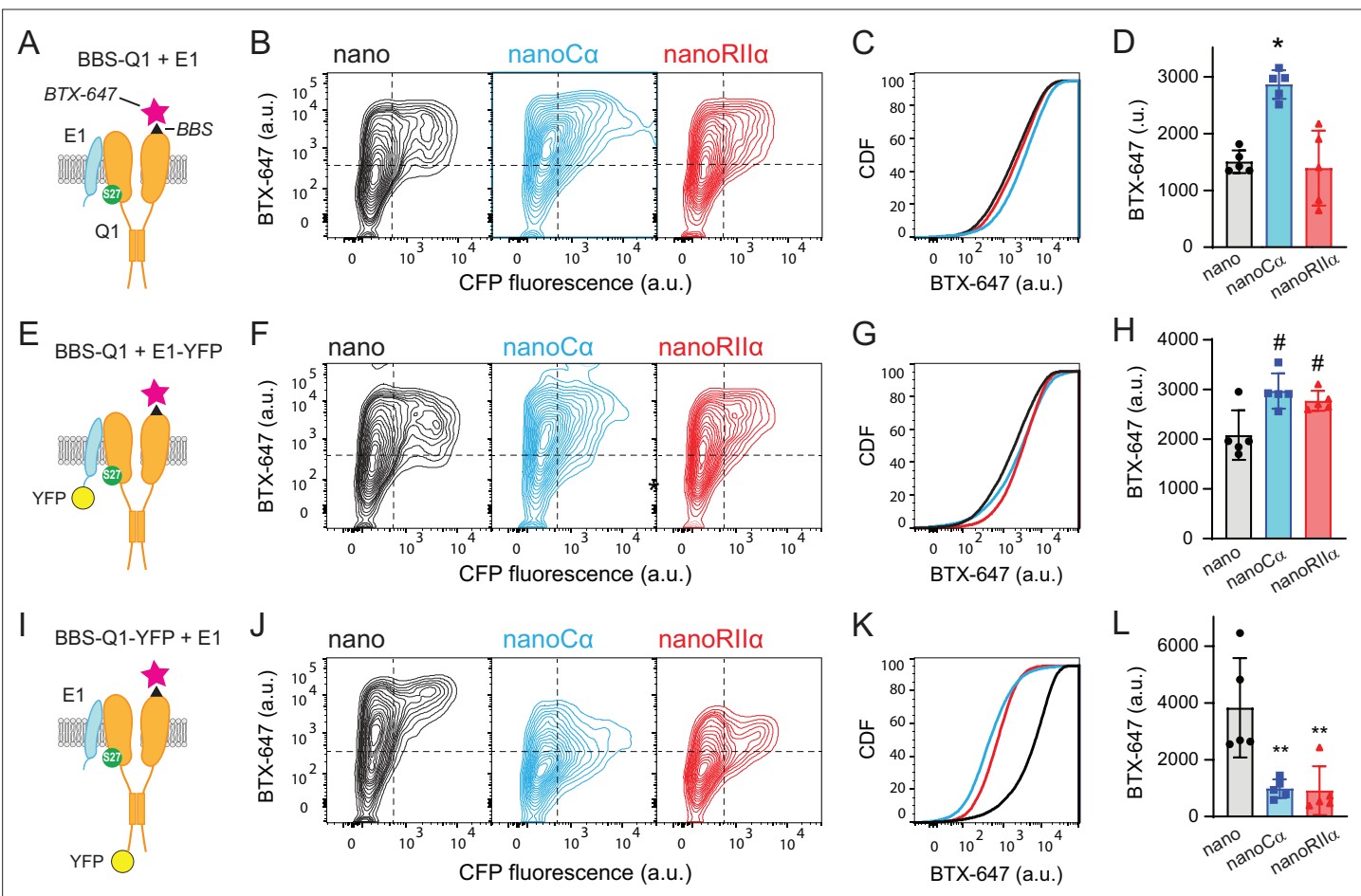

**Figure 3.** Tethering Cα and RIIα to either E1 or Q1 yields differential effects on channel surface density. (**A**) Cartoon showing strategy for surface labeling of BBS-Q1/E1 using BTX-647. (**B**) Flow cytometry contour plots showing surface channels (BTX-647 fluorescence) and nano expression (CFP fluorescence) in cells expressing BBS-Q1/E1 with nano (*left*), nanoCα (*middle*), or nanoRIIα (*right*). (**C**) Corresponding cumulative distribution (CDF) histograms of BTX-647 fluorescence. Plot generated from population of CFP-positive cells. (**D**) Channel surface density (mean BTX-647 fluorescence in CFP-positive cells). *p=0.0003, one-way ANOVA and Tukey HSD post hoc test. (**E–H**) Cartoon, contour plots, CDF, and average surface labeling of BBS-Q1 in cells expressing BBS-Q1/E1-YFP with nano, nanoCα, or nanoRIIα, same format as A–D. #p<0.05, one-way ANOVA and Tukey HSD post hoc test. (**I–L**) Cartoon, contour plots, CDF, and normalized average surface labeling of BBS-Q1-YFP in cells expressing BBS-Q1-YFP/E1 with nano, nanoCα, or nanoRIIα, same format as A–D. **p<0.05, one-way ANOVA and Tukey HSD post hoc test.

The online version of this article includes the following source data for figure 3:

**Source data 1.** Tethering Cα and RIIα to either E1 or Q1 yields differential effects on channel surface density.

to control (*Figure 2—figure supplement 2*). Thus, nanoCα does not indiscriminately inhibit currents when recruited to the C-terminus of $K^+$ channel pore-forming subunits.

## Recruiting Cα or RIIα to either E1 or Q1 yields distinctive effects on channel trafficking

To distinguish the mechanisms underlying the differential functional impact of nanoRIIα and nanoCα on $I_{Ks}$ depending on whether they are targeted to either E1 or Q1, we assessed their effects on channel trafficking to the cell surface under different conditions. We used a Q1 construct incorporating a high-affinity bungarotoxin binding site (BBS) in the extracellular S1-S2 loop to permit detection of channels at the surface in non-permeabilized cells with Alexa Fluor-647 conjugated α-bungarotoxin (BTX-647) (*Aromolaran et al., 2014*; *Kanner et al., 2017*; *Figure 3A*). Control cells expressing BBS-Q1+E1+nano (in a P2A-CFP construct) displayed robust surface fluorescence as indicated by flow cytometry detection of BTX-647 and CFP (marker for nano expression) fluorescence signals. Co-expressing nanoCα resulted in an elevated BTX-647 fluorescence signal (mean$_{BTX-647}$=1505.6 ± 79.88 a.u., N=5 for BBS-Q1+E1+nano; and mean$_{BTX-647}$=2864.2 ± 101.44 a.u., N=5 for BBS-Q1+E1+nanoCα; p<0.001, one-way ANOVA and Tukey HSD post hoc test), indicating an increase in channel surface density, while nanoRIIα had no significant impact (mean$_{BTX-647}$=1393 ± 263.82 a.u., N=5 for BBS-Q1+E1+nanoRIIα) (*Figure 3A–D*). Next, we examined how tethering of nano, nanoCα, or nanoRIIα to E1-YFP affected channel trafficking. In this configuration, both nanoCα and nanoRIIα resulted in an increase in BTX-647 fluorescence compared to nano control (mean$_{BTX-647}$=2083.2 ± 199.60 a.u., N=5 for BBS-Q1+E1-YFP+nano; and mean$_{BTX-647}$=2970 ± 142.26 a.u., N=5 for BBS-Q1+E1-YFP+nanoCα; mean$_{BTX-647}$=2772.6 ± 80.99 a.u., N=5 for BBS-Q1+E1-YFP+nanoRIIα, p<0.01, one-way ANOVA and Tukey HSD post hoc test) (*Figure 3E–H*). Finally, we examined the impact of individually targeting nano, nanoCα, and nanoRIIα directly to Q1-YFP (*Figure 3I*). Both nanoCα and nanoRIIα led to a dramatic decrease in BTX-647 fluorescence compared to nano control (mean$_{BTX-647}$=3830.4 ± 700.03 a.u., N=5 for BBS-Q1-YFP+E1+nano; mean$_{BTX-647}$=908 ± 132.95 a.u., N=5 for BBS-Q1-YFP+E1+nanoCα; and mean$_{BTX-647}$=910 ± 344.68 a.u., N=5 for BBS-Q1-YFP+E1+nanoRIIα, p<0.001, one-way ANOVA and Tukey HSD post hoc test) (*Figure 3J–L*), indicating that recruiting these PKA subunits to Q1 C-terminus suppresses channel surface density, and rationalizing the inhibitory impact on $I_{Ks}$.

## Targeting Cα or RIIα to Q1 retains channels in the ER and Golgi

Given that targeting either Cα or RIIα specifically to Q1 impairs channel surface trafficking (*Figure 3G–I*), we hypothesized that the channels remained trapped in intracellular compartments. To determine intracellular localization of the channels, we used confocal imaging to examine the subcellular localization of Q1-YFP when co-expressed with nano, nanoCα, or nanoRIIα. We co-expressed either mCherry-tagged ER- or Golgi-localizing marker proteins, respectively, under the different experimental conditions. When co-expressed with nano, Q1-YFP showed some fraction present in both the ER (*Figure 4A*, top row) and Golgi (*Figure 4B*, top row) compartments, but also clear staining at the surface membrane. By contrast, co-expressing either nanoCα or nanoRIIα significantly increased the co-localization of Q1-YFP with ER-mCherry (Pearson's co-localization coefficient [PCC]=0.85 ± 0.020, n=9 for nano; PCC = 0.90 ± 0.015, n=7 for nanoCα; and PCC = 0.94 ± 0.010, n=8 for nanoRIIα; p=0.0049, one-way ANOVA and Tukey HSD post hoc test) (*Figure 4A and B*) and Golgi-mCherry (PCC = 0.84 ± 0.016, n=9 for nano; PCC = 0.90 ± 0.012, n=8 for nanoCα; and PCC = 0.90 ± 0.011, n=6 for nanoRIIα; p=0.0065, one-way ANOVA and Tukey HSD post hoc test) (*Figure 4C and D*), respectively, with no discernible YFP signal at the cell surface. Thus, targeting Cα or RIIα to Q1-YFP leads to increased retention of the channel in the ER and Golgi compartments.

## Inducible recruitment of nanoCα to Q1 reveals slow temporal regulation of channel trafficking

We wondered about the kinetics of this newly revealed regulation of channel trafficking brought on by tethering Cα or RIIα to Q1 C-terminus. To gain insights into this question, we exploited a small molecule-induced heterodimerization strategy, in which rapamycin simultaneously binds to FK506 binding protein (FKBP) and the FKBP-rapamycin binding domain (FRB) of the mammalian target of rapamycin (*Crabtree and Schreiber, 1996*; *Inoue et al., 2005*) to enable acute temporal control of nanoCα recruitment to Q1 C-terminus (*Figure 5A*). We fused anti-GFP/YFP nanobody to FKBP and

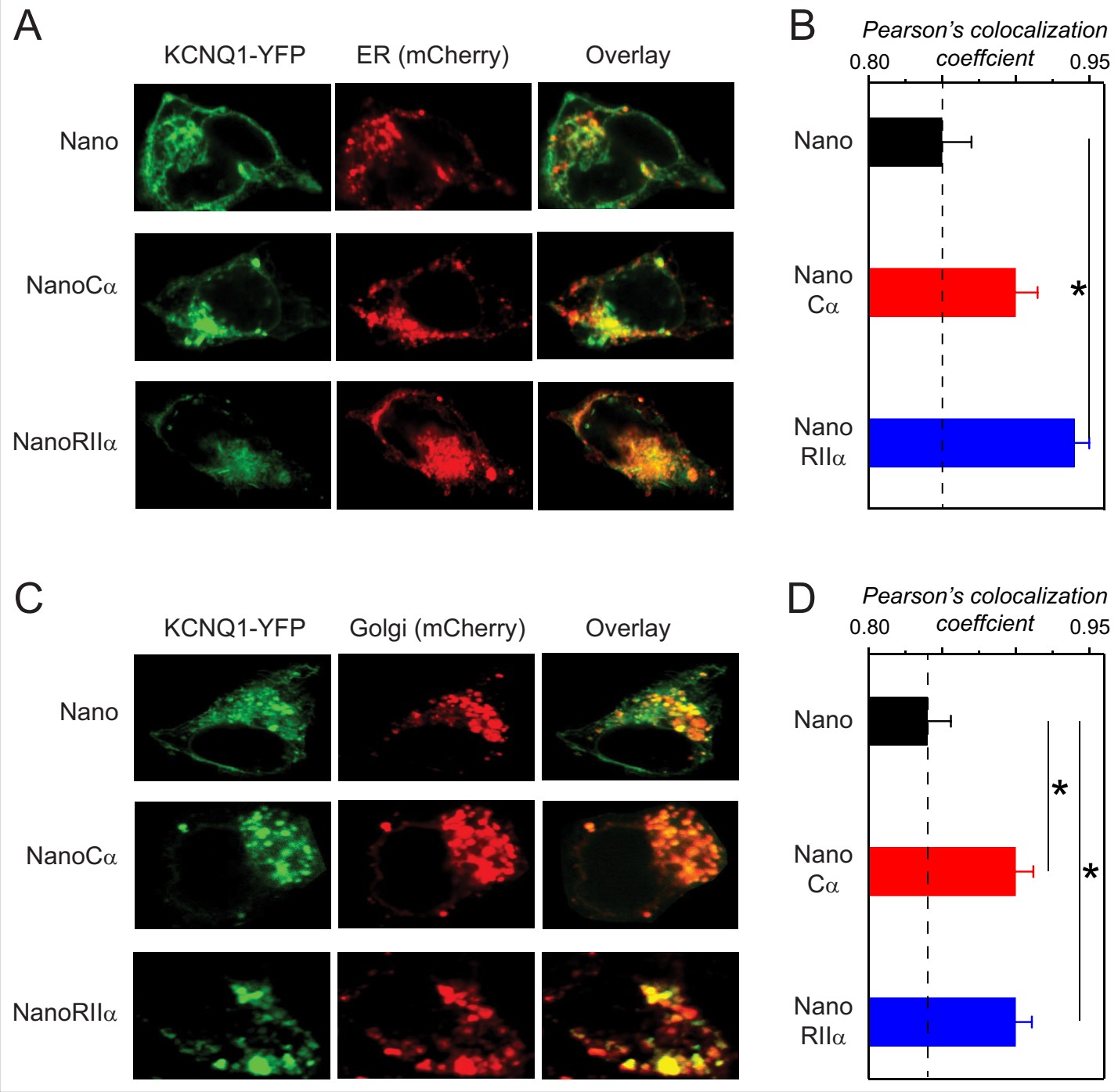

**Figure 4.** Subcellular localization of KCNQ1 tethered to nano, nanoCα, or nanoRIIα. (**A**) Representative confocal images of HEK293 cells expressing Q1-YFP/E1 and ER-mCherry marker with nano, nanoCα, or nanoRIIα. (**B**) Co-localization of Q1-YFP with ER-mCherry assessed by Pearson's co-localization coefficient; $n=9$ for nano, $n=7$ for nanoCα and $n=8$ for nanoRIIα. (**C**) Representative confocal images of HEK293 cells expressing Q1-YFP/E1 and Golgi-mCherry marker with nano, nanoCα, or nanoRIIα. (**D**) Co-localization of Q1-YFP with ER-mCherry assessed by Pearson's co-localization coefficient; $n=9$ for nano, $n=8$ for nanoCα, and $n=6$ for nanoRIIα. *$p<0.05$, one-way ANOVA and Tukey HSD post hoc test.

Cα to FRB and generated an FKBPnano-P2A-FRBCα construct to ensure 1:1 expression of FKBPnano and FRBCα as separate proteins. We transfected HEK293 cells with BBS-Q1-YFP+E1+FKBPnano-P2A-FRBCα and monitored channel surface density at various time points after adding 1 μM rapamycin to heterodimerize FKBP and FRB fusion proteins (*Figure 5A and B*). Channel surface density was relatively unchanged from control (pre-rapamycin) at the 20 and 60 min time points post-rapamycin

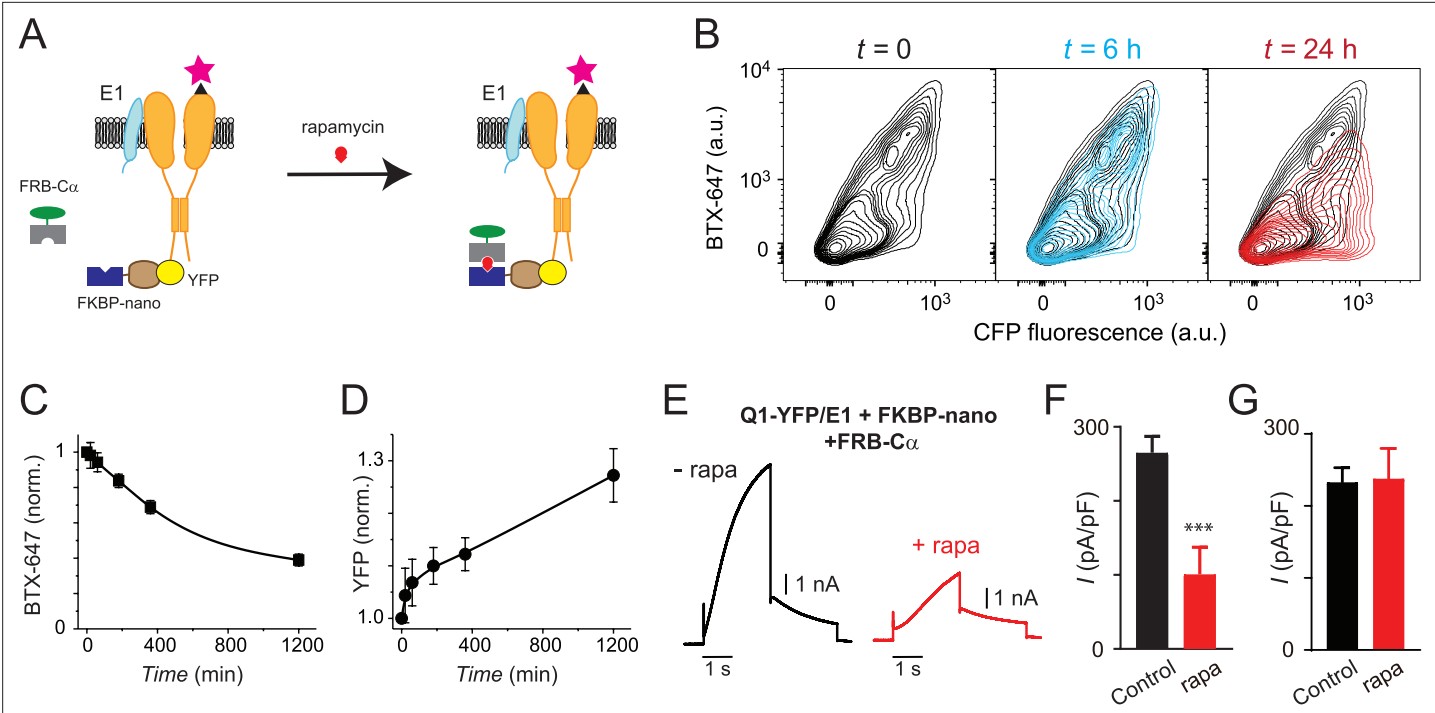

**Figure 5.** Slow temporal regulation of channel trafficking by targeted induced recruitment of nanoCα to Q1 C-terminus. (A) Cartoon of FK506 binding protein (FKBP)/FKBP-rapamycin binding domain (FRB) heterodimerization strategy utilized for rapamycin-induced recruitment of engineered Cα to BBS-Q1-YFP/E1. (B) Exemplar flow cytometry contour plots showing surface expression (BTX-647 fluorescence) and CFP fluorescence in cells expressing BBS-Q1-YFP/E1 with FRB-Cα and FKBP-nano at times $t$=0 (*left*), $t$=6 hr (*middle*), and $t$=24 hr (*right*) after rapamycin addition. (C) Normalized mean Q1 surface density (BTX-647 fluorescence) plotted as a function of time after rapamycin induction. (D) Normalized mean Q1 total expression (YFP fluorescence) plotted as a function of time after rapamycin induction. (E) Exemplar $I_{Ks}$ traces recorded in Chinese hamster ovary (CHO) cells co-expressing KCNQ1-YFP/KCNE1/nano-FKBP-FRB-Cα incubated 20 hr either without (*left*) or with (*right*) rapamycin. (F) Mean current densities in CHO cells co-expressing KCNQ1-YFP/KCNE1/nano-FKBP-FRB-Cα without rapamycin (black, $n$=10) or after 20 hr rapamycin incubation (red, $n$=14). ***$p$<0.001, paired $t$ test. (G) Mean current densities in control cells co-expressing KCNQ1-YFP/KCNE1 without rapamycin (black, $n$=8) or after 20 hr rapamycin incubation (red, $n$=9).

The online version of this article includes the following source data for figure 5:

**Source data 1.** Slow temporal regulation of channel trafficking by targeted induced recruitment of nanoCα to Q1 C-terminus.

(*Figure 5C*). Beyond that time, Q1-YFP surface density steadily declined, reaching 40% of control 20 hr after rapamycin. Concomitant with the time-dependent decrease in channel surface density post-rapamycin, there was a reciprocal increase in YFP fluorescence, suggesting a stabilization of the total Q1 protein (*Figure 5D*). Consistent with the induced decrease in Q1 surface density, whole-cell patch clamp showed that in cells expressing Q1-YFP+E1+FKBPnano-P2A-FRB-Cα treatment with 1 μM rapamycin for 16 hr resulted in a 60% decline in $I_{Ks}$ density (*Figure 5E and F*). In control experiments, rapamycin had no impact on currents from cells expressing Q1-YFP+E1 alone (*Figure 5G*).

## Proteomic analysis reveals distinctive phosphorylation of Q1 residues by Cα targeted to Q1-YFP compared to free Cα

The most parsimonious explanation for the divergent functional effects of Cα either targeted to Q1 or expressed free (or targeted to E1) is that the two treatments result in the phosphorylation of distinct complements of Ser and/or Thr residues on Q1. We used liquid chromatography with tandem mass spectrometry (LC-MS/MS) to identify residues on Q1-YFP that are phosphorylated when the channel is co-expressed with nano (basal control), nanoCα, or free Cα. Altogether, we identified 19 Ser and Thr residues that were phosphorylated under at least one of the three experimental conditions (*Figure 6A* and *Figure 6—figure supplement 1*). The pattern of modification of these residues fell in one of three categories: (1) basally phosphorylated with no increase with either nanoCα or free Cα (S6, S402, S407, and S409); (2) low or undetectable basal phosphorylation with increases in modification seen

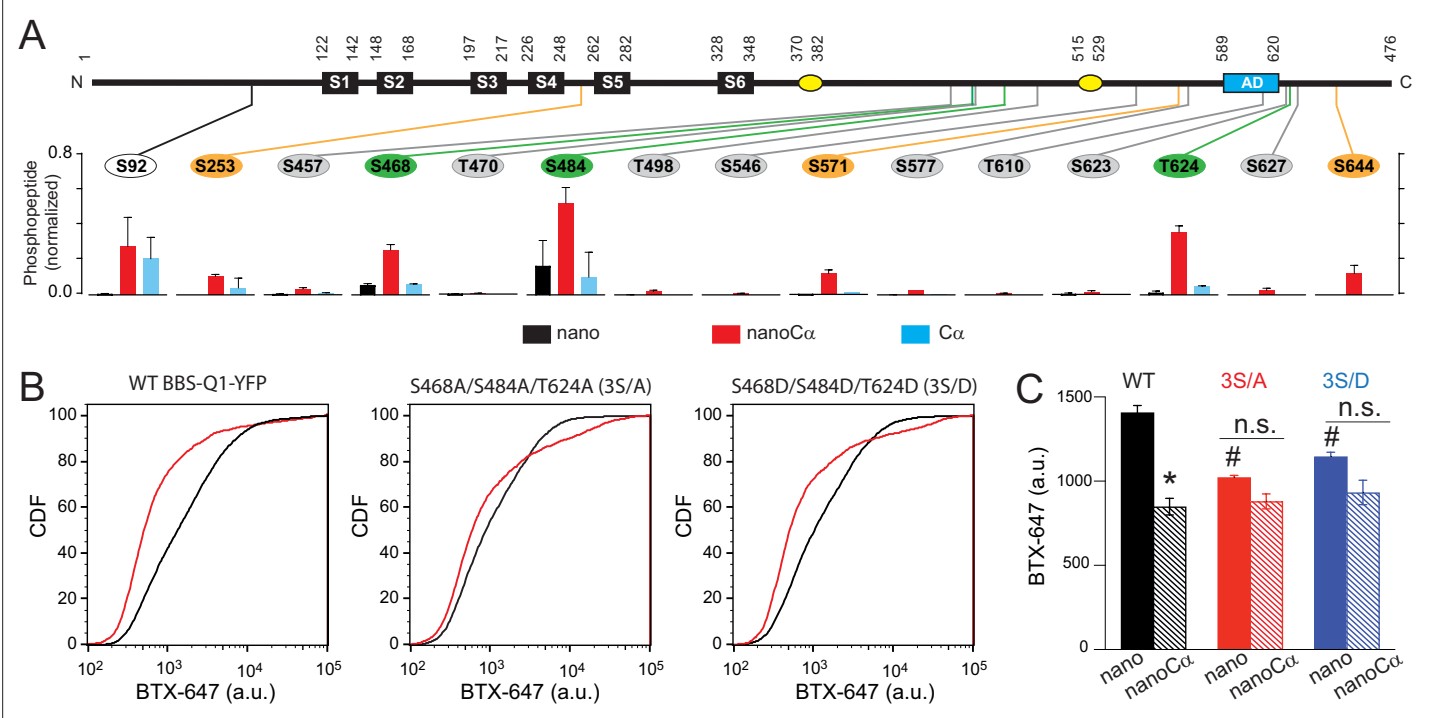

**Figure 6.** Potential phosphorylation sites involved in protein kinase A (PKA) modulation of KCNQ1 trafficking. (A) *Top*, schematic of Q1 showing positions of Ser and Thr residues where phosphorylation was increased when nanoCα was targeted to Q1 C-terminus. *Bottom*, relative abundance of phosphorylated KCNQ1-YFP peptides identified using mass spectrometry in cells co-expressing nano (black), nanoCα (red), or free Cα (cyan). (B) Exemplar CDF plots showing channel surface density in cells expressing WT BBS-Q1-YFP (*left*), BBS-3S/A-YFP (*middle*), or BBS-3S/D-YFP (*right*) in the absence (black traces) or presence (red traces) of nanoCα. (C) Channel surface density (mean BTX-647 fluorescence in YFP-positive cells) in cells expressing WT BBS-Q1-YFP, BBS-3S/A-YFP, or BBS-3S/D-YFP in the presence of either nano or nanoCα. WT BBS-Q1-YFP (nano, *N*=4; nanoCα, *N*=4; *p<0.001, unpaired t-test). BBS-3S/A-YFP (nano, *N*=4; nanoCα, *N*=4; p=0.063, unpaired t-test). BBS-3S/D-YFP (nano, *N*=4; nanoCα, *N*=4; p=0.079, unpaired t-test). #p<0.001 compared to WT+nano, one-way ANOVA and Tukey HSD post hoc test.

The online version of this article includes the following source data and figure supplement(s) for figure 6:

**Source data 1.** Potential phosphorylation sites involved in protein kinase A (PKA) modulation of KCNQ1 trafficking.

**Source data 2.** Potential phosphorylation sites involved in protein kinase A (PKA) modulation of KCNQ1 trafficking.

**Figure supplement 1.** Relative abundance of phosphorylated KCNQ1-YFP peptides identified using mass spectrometry in cells co-expressing nano (black), nanoCα (red), or free Cα (cyan).

**Figure supplement 1—source data 1.** Relative abundance of phosphorylated KCNQ1-YFP peptides identified using mass spectrometry in cells co-expressing nano, nanoCα, or free Cα.

with both nanoCα and free Cα co-expression (S92 and S253); and (3) low or undetectable basal phosphorylation with an increase in modification seen only with nanoCα co-expression (S457, S468, T470, S484, T498, S546, S571, S577, T610, S623, T624, S627, and S644) (*Figure 6—figure supplement 1*).

Given the strong impact of nano-Cα on tetrameric Q1 trafficking, we deduced that the residue(s) important for the functional effect would present as a substantial fraction (>25%) of phosphorylated peptide to total peptide ratio with nano-Cα treatment. Normalization of the fraction of phosphorylated peptide suggested three residues of potential interest – (S468, S484, and T624) (*Figure 6A*). We tested the impact of mutating these three residues to either alanines (3S/A) or phosphomimetic aspartates (3S/D) on baseline trafficking of Q1-YFP and on nanoCα-mediated decrease in channel surface density (*Figure 6B, C*). In WT BBS-Q1-YFP, nanoCα produced a 40% decrease in channel surface density (mean$_{BTX-647}$=1408.25 ± 40.57 a.u., *N*=4 for WT BBS-Q1-YFP+nano; and mean$_{BTX-647}$=849 ± 49.57 a.u., *N*=4 for WT BBS-Q1-YFP+nanoCα; p=0.0003, unpaired t-test) (*Figure 6B, C*). Compared to WT Q1-YFP, 3S-A showed a 27% decrease in the baseline channel surface expression and a substantially reduced response to nanoCα (mean$_{BTX-647}$=1023.5 ± 10.99 a.u., *N*=4 for BBS-Q1[3S/A]-YFP+nano; and mean$_{BTX-647}$=880 ± 44.15 a.u., *N*=4 for BBS-Q1[3S/A]-YFP+nanoCα; p=0.063, unpaired t-test) (*Figure 6B and C*). The diminished effect of nanoCα on S3-A suggests that increased phosphorylation

of at least one of the three residues plays a significant role in the impact of nanoCα on WT Q1 surface density. However, the unexpected decreased baseline surface expression of 3S-A suggests a potential positive effect of phosphorylation of at least one of these residues on basal surface density. Consistent with this interpretation, 3S-D showed an intermediate effect with a 19% decrease in baseline surface density compared to WT, and a diminished response to nanoCα (mean$_{BTX-647}$=1023.5 ± 10.99 a.u., $N$=4 for BBS-Q1[3S/D]-YFP+nano; and mean$_{BTX-647}$=880 ± 44.15 a.u., $N$=4 for BBS-Q1[3S/D]-YFP+nanoCα; p=0.063, unpaired t-test) (*Figure 6B and C*). Overall, we conclude that phosphorylation of multiple Ser/Thr residues underlie the large impact of nanoCα targeted to the Q1 C-terminus on channel surface density. Moreover, our results suggest that phosphorylation of distinct Ser/Thr residues may have either a positive or negative impact on channel trafficking, and channel surface density will be determined by a combinatorial contribution of these negative and positive influences.

## Discussion

PKA is known to phosphorylate over 100 distinct substrates in cells, altering protein function to powerfully regulate physiology (*Shabb, 2001*). In the heart, β-adrenergic agonist activation of PKA results in the increase in contractility and heart rate that underlies the fight-or-flight response. PKA activation increases cardiac L-type currents ($I_{Ca,L}$) by phosphorylating the small G-protein Rad which constitutively inhibits a sub-population of Ca$_V$1.2 channels in cardiomyocytes (*Liu et al., 2020*; *Finlin et al., 2003*). Phosphorylation of residues in the Rad C-terminus causes it to disengage from the channel, resulting in augmented $I_{Ca,L}$ that is critical for the positive inotropic response (*Liu et al., 2020*; *Papa et al., 2022*). If unopposed, the PKA-mediated elevated inward $I_{Ca,L}$ would result in a prolongation of the APD during exercise or fright (*Volders et al., 2003*; *Gadsby, 1983*). This would have two adverse consequences. First, a diminished diastolic period would result in inadequate filling of the heart during diastole, particularly in light of the increased heart rate. Second, the prolonged APD would increase the propensity for lethal cardiac arrhythmias. These injurious sequelae are normally prevented because PKA also increases the amplitude and slows deactivation of $I_{Ks}$, providing a countercurrent that serves to maintain a physiologically optimum APD (*Banyasz et al., 2014*; *Volders et al., 2003*; *Gadsby, 1983*). PKA regulation of $I_{Ks}$ critically depends on the scaffold protein AKAP9 which binds KCNQ1 and anchors the PKA holoenzyme in proximity to the channel (*Marx et al., 2002*). Loss-of-function mutations in KCNQ1 cause LQT1, characterized by a prolonged APD and increased susceptibility to exertion triggered lethal cardiac arrhythmias. Some LQT1 mutations occur on the binding interface with AKAP9, and thus render the channel unable to be regulated in response to β-adrenergic receptor activation (*Marx et al., 2002*). Further, human mutations in AKAP9 that inhibit the interaction with KCNQ1 have been proposed to cause LQT11 (*Schwartz et al., 2012*; *Chen et al., 2007*). We show here that the scaffolding function of AKAP9 that enables acute PKA regulation of $I_{Ks}$ can be supplanted by a nanobody fused to RIIα and targeted to E1 in the channel complex. This result suggests a novel approach to develop treatments for LQTS cases arising from disruption of Q1/AKAP9 molecular and/or functional interaction – bivalent small molecules that induce proximity of RIIα/RIIβ (or RI) to E1. This approach would require the generation of a nanobody or other antibody-mimetic that binds to the intracellular domain of E1. Beyond $I_{Ks}$, diverse AKAPs play an essential role in the proper organization and restriction of PKA signaling to distinct proteins and organelles, and disruption of this capacity leads to serious pathologies (*Wong and Scott, 2004*; *Kjällquist et al., 2018*; *Gold et al., 2013*). Our results suggest a bioengineering approach to rectify such aberrant PKA signaling even in the case of a malfunctioning AKAP.

We surprisingly found that nanoRIIα and nanoCα targeted to the C-terminus of Q1 yielded a qualitatively different outcome compared to when they were recruited to E1 C-terminus. The contrasting results illustrate both the challenges and opportunities likely to be encountered in adapting targeted induced proximity of kinases. By contrast with the binary outcomes achieved with the more established TPD with PROTACS (*Sakamoto et al., 2001*; *Békés et al., 2022*), and more recently TPS (*Kanner et al., 2020*; *Henning et al., 2022*), PHICS are likely to yield more diverse and nuanced outcomes, in part because the complement of residues modified may depend on the targeting site. The nanobody-based method we have outlined here offers a way to establish the rules that enable a more rational approach to the design and development of PHICS with predicted outcomes. The finding that nanoRIIα directed to Q1 C-terminus constitutively inhibited channel trafficking indicates that Cα is recruited to the channel complex but not absolutely held in an inactive state under basal

conditions. This is most likely due to basal levels of cAMP in the cells being sufficient to bind to nano-RIIα tethered to Q1 C-terminus and thereby partially activate PKA to an extent where it can chronically affect channel trafficking. In this configuration, the Cα need not dissociate from nanoRIIα in order to phosphorylate Q1 (*Smith et al., 2017*).

Is the dramatic decrease in channel trafficking and $I_{Ks}$ induced by nanoCα targeted to Q1 a de novo functional property, or is it an amplified version of a naturally occurring phosphate-switch mechanism that controls Q1 surface density? To this point, proteomics analyses indicated that several Ser/Thr residues are basally phosphorylated and that this modification is selectively potentiated by nanoCα targeted to Q1, in contrast to free Cα expression. Interestingly, examination of the subcellular localization of Q1-YFP indicates expression of the channel in the ER, Golgi, and the plasma membrane. Distinctive regulation of Q1 surface density by kinases has been reported in several previous studies. Activation of protein kinase C (PKC) with phorbol 12-myristate 13-acetate decreased $I_{Ks}$ density by >60% within 30 min in transfected CHO cells due to Q1/E1 internalization that was dependent on E1 residue S102 (*Kanda et al., 2011*). Stimulation of $\alpha_1$ adrenergic receptors produces an acute inhibition of Q1-alone currents (60% reduction in 30 min) mediated by an internalization of the channel (*Kurakami et al., 2019*). The mechanism was proposed to involve α1 adrenergic receptor activation of AMPK which subsequently stimulates Nedd4-2 to increase Q1 ubiquitination and internalization (*Kurakami et al., 2019*). By comparison with these previously reported forms of kinase-mediated inhibition of Q1 surface density, the mechanism described here is clearly distinct, having a slower time course, and being independent of either E1 expression or Nedd4-2 involvement. The characteristics of the targeted nanoCα/nanoRIIα-mediated inhibition of Q1 trafficking are most akin to a reported chronic down-regulation of $I_{Ks}$ attributed to a PKCε-induced reduction in Q1 forward trafficking (*Gou et al., 2021*). We speculate that the chronic targeting of PKA to Q1 C-terminus amplifies a physiological phosphorylation-controlled forward trafficking gate that is normally regulated by another kinase(s) such as, potentially, PKCε.

It is worth noting that there are aspects of AKAP function on $I_{Ks}$ that we would not expect to be reconstituted with the current configuration of nanoRIIα. Beyond acting as a mere scaffold, binding of AKAP9 to Q1 itself has been shown to modulate channel gating downstream of PKA phosphorylation (*Kurokawa et al., 2004*). Further, in addition to anchoring PKA, AKAPs typically serve as hubs that tether other enzymes including phosphatases and phosphodiesterases that influence spatiotemporal aspects of kinase action on substrates (*Wong and Scott, 2004*). AKAP9 binds to the Q1 C-terminus and would thus tether the PKA holoenzyme to this site in the channel complex (*Marx et al., 2002*). Yet, Q1/E1/AKAP9 complexes are not absolutely retained intracellularly, unlike what we observe for nanoCα, and to a lesser extent for nanoRIIα, targeted to Q1-YFP. One possibility to account for the apparent discrepancy is that a phosphatase anchored near Q1 by AKAP9 rapidly dephosphorylates residues that when phosphorylated result in channel retention. Alternatively, the relatively large size of AKAP9 (453 kDa) could position the PKA holoenzyme in a geometrically unfavorable configuration to phosphorylate residues involved in the channel intracellular retention response.

Overall, our findings advance the notion of targeted phosphorylation by recruitment of kinases as a versatile mechanism to control protein function as potent research tools or potential therapeutics. Exploring the numerous potential applications of this iteration of TIPE technology is an interesting prospect for future experiments.

# Materials and methods

**Key resources table**

| Reagent type (species) or resource | Designation | Source or reference | Identifiers | Additional information |
|---|---|---|---|---|
| Cell line (*Homo sapiens*) | HEK293 | ATCC | RRID:CVCL_0045 | Laboratory of Dr. Robert Kass |
| Cell line (*Homo sapiens*) | CHO | ATCC | RRID:CVCL_0214 | CHO-K1, ATCC, CCL-61 |
| Antibody | Anti-Q1 (Rabbit polyclonal) | Alomone | RRID:AB_2040099 | IP (1:1000), WB (1:1000) |
| Antibody | Anti-PKA (Rabbit monoclonal) | Abcam | Cat# ab76238, RRID:AB_1523259 | IP(1:1000) WB (1:1000) |

*Continued on next page*

*Continued*

| Reagent type (species) or resource | Designation | Source or reference | Identifiers | Additional information |
|---|---|---|---|---|
| Antibody | Anti-actin (Rabbit polyclonal) | Abcam | Cat# ab197345 | WB (1:2000) |
| Antibody | Anti-pQ1 (Rabbit polyclonal) | PMID:12566567 | | WB (1:250) |
| Recombinant DNA reagent | BBS-Q1-YFP (plasmid) | PMID:25344363 | | |
| Recombinant DNA reagent | BBS-Q1 (plasmid) | PMID:25344363 | | |
| Recombinant DNA reagent | Q1-YFP (plasmid) | PMID:25344363 | | |
| Recombinant DNA reagent | Q1 (plasmid) | PMID:25344363 | | From the lab of William Kobertz |
| Recombinant DNA reagent | E1-YFP (plasmid) | PMID:25344363 | | |
| Recombinant DNA reagent | E1 (plasmid) | PMID:25344363 | | From the lab of William Kobertz |
| Recombinant DNA reagent | Yotiao (plasmid) | PMID:15528278 | | |
| Recombinant DNA reagent | Q1[S27A]-YFP | This paper | | Made by site-directed mutagenesis; see Plasmid constructs and mutagenesis |
| Recombinant DNA reagent | NanoCα-P2A-CFP (plasmid) | This paper | | Made by gene synthesis (Genewiz) and cloning; see Plasmid constructs and mutagenesis |
| Recombinant DNA reagent | NanoCα[T198A]-P2A-CFP (plasmid) | This paper | | Made by site-directed mutagenesis; see Plasmid constructs and mutagenesis |
| Recombinant DNA reagent | Cα-P2A-CFP (plasmid) | This paper | | Made by gene synthesis (Genewiz) and cloning; see Plasmid constructs and mutagenesis |
| Recombinant DNA reagent | NanoRIIα-P2A-CFP (plasmid) | This paper | | Made by gene synthesis (Genewiz) and cloning; see Plasmid constructs and mutagenesis |
| Recombinant DNA reagent | Nano-P2A-CFP (plasmid) | PMID:29256394 | | |
| Peptide, recombinant protein | Protein A/G Sepharose beads | Rockland | Cat# PAG50-00-0002 | |
| Peptide, recombinant protein | α-Bungarotoxin, Alexa Fluor 647 conjugate | Thermo Fisher scientific | Cat# B35450 | |
| Commercial assay or kit | Quik-Change Site-Directed Mutagenesis Kit | Agilent Technologies | Cat# 200523 | |
| Chemical compound, drug | Rapamycin | Sigma | Cat# 553211-1MG | |
| Software, algorithm | FlowJo | FlowJo, LLC | RRID:SCR_008520 | |
| Software, algorithm | GraphPad Prism | GraphPad Software Inc | RRID:SCR_002798 | |
| Software, algorithm | Origin | OriginLab Corporation | RRID:SCR_014212 | |
| Software, algorithm | PulseFit | HEKA | | |

## Plasmid constructs and mutagenesis

Human Q1, Q1[S27A], E1, Cα, and yotiao were cloned in pcDNA3.1 (+) vector. Q1-YFP, E1-YFP, and Q1-BBS-YFP were made as previously described (*Aromolaran et al., 2014*). NanoCα and nanoRIIα

were created by gene synthesis (Genewiz), and featured the coding sequence for GFP nanobody (vhhGFP4) (*Kubala et al., 2010*) in frame with cDNA for PKA Cα (NM_002730) and RIIα (X14968) subunits, respectively. Cα, nanoCα, nanoRIIα fragments were amplified using the polymerase chain reaction and cloned into a customized bicistronic expression vector (xx-P2A-CFP) (*Kanner et al., 2017*). Q1[S27A] and nanoCα[T198A] mutations were generated using the Quik-Change Lightning Site-Directed Mutagenesis Kit (Agilent Technologies, Santa Clara, CA, USA). All constructs were verified by sequencing.

## Cell culture and transfection

Low-passage-number Chinese hamster ovary (CHO-K1) cells (American Type Culture Collection) were cultured at 37°C in Ham's F12 medium with 10% fetal bovine serum (FBS) and 100 µg/mL of penicillin-streptomycin. Cells were transiently transfected with desired plasmids including Q1, Q1[S27A], Q1-YFP, E1, E1-YFP, Cα, nanoCα, nanoRIIα, nanoCα[T198A], and yotiao in 25 cm² flask. HEK 293 (RRID:CVCL_0045) cells were maintained in DMEM medium with 10% FBS and 100 µg/mL of penicillin-streptomycin. HEK 293 cells and CHO cells were transiently transfected with desired plasmids for western blot or flow cytometry. Lipofectamine and Plus reagent (Invitrogen) were used for transfection. The cell lines used have been authenticated by STR profiling and determined to be mycoplasma-free using the MycoFluor Mycoplasma Detection Kit (Invitrogen, Carlsbad, CA, USA).

## Western blot

CHO cells cultured in 35 mm dishes were transfected with designed plasmids, such as Q1, E1-YFP, Cα, and nanoCα (total DNA is 2.5 µg). Two days after transfection, cells were lysed in a lysis buffer (150 mM $NaCl_2$, 1 mM EDTA, 10 mM Tris, 1% Triton X-100, pH 7.5). Cell lysates were resolved by 4–20% SDS-PAGE. Phosphorylated Q1 channels were detected by using the rabbit anti-phosphoQ1 antibody (1:250) and visualized by chemiluminescence with the ECL-plus western blotting detection system (Amersham Pharmacia). Rabbit anti-Q1 antibody (1:1000, Alomone labs, Israel) and rabbit anti-actin antibody (1:2000, Abcam, USA) were used to detect total Q1 channels or actin protein in the lysate.

## Electrophysiology

Cells were plated in 3.5 cm culture dishes on the stage of an inverted microscope (OLYMPUS BH2-HLSH, Precision Micro Inc, Massapequa, NY, USA). Currents were recorded at room temperature (RT) using the whole-cell patch clamp technique by an Axopatch 200B amplifier (Axon Instruments, Foster City, CA, USA). Patch clamp protocols have been described previously (*Terrenoire et al., 2009*). Briefly, after 500 ms of holding potential at –70 mV, the voltage was stepped to +60 mV for 2 s and then followed by 2 s repolarizing pulses to –40 mV during which $I_{Ks}$ tail current was measured (stimulation frequency was 0.06 Hz). External solution contained the following: 132 mM NaCl, 4.8 mM KCl, 2 mM $CaCl_2$, 1.2 mM $MgCl_2$, 10 mM HEPES, and 5 mM glucose (pH was adjusted to 7.4 with NaOH). Internal solution contained the following: 110 mM KCl, 5 mM ATP-$K_2$, 11 mM EGTA, 10 mM HEPES, 1 mM $CaCl_2$, and 1 mM $MgCl_2$ (pH was adjusted to 7.3 with KOH). Pipette series resistance was typically 1.5–3 MΩ when filled with internal solution. Currents were sampled at 10 kHz and filtered at 5 kHz. Traces were acquired at a repetition interval of 10 s.

## Flow cytometry assay

Cell surface and total ion channel pools were assayed by flow cytometry in live, transfected HEK293 cells as previously described (*Aromolaran et al., 2014*; *Kanner et al., 2017*; *Kanner et al., 2018*). Briefly, 48 hr post-transfection, cells cultured in 12-well plates were gently washed with ice-cold PBS containing $Ca^{2+}$ and $Mg^{2+}$ (in mM: 0.9 $CaCl_2$, 0.49 $MgCl_2$, pH 7.4), and then incubated for 30 min in blocking medium (DMEM with 3% BSA) at 4°C. The cells were then incubated with 1 µM Alexa Fluor 647-conjugated α-bungarotoxin (BTX-647; Life Technologies) in DMEM/3% BSA on a rocker at 4°C for 1 hr, followed by washing three times with PBS (containing $Ca^{2+}$ and $Mg^{2+}$). Cells were gently harvested in $Ca^{2+}$-free PBS, and assayed by flow cytometry using a BD LSRII Cell Analyzer (BD Biosciences, San Jose, CA, USA). CFP- and YFP-tagged proteins were excited at 407 and 488 nm, respectively, and Alexa Fluor 647 was excited at 633 nm.

## Confocal imaging

HEK293 cells transfected with enhanced yellow fluorescent protein-tagged KCNQ1 and subcellular marker proteins for ER or Golgi (mCherry-tagged) with either nano, nanoCα, or nanoRIIα with CFP marker were analyzed 24–48 hr after transfection using an inverted Nikon Eclipse Ti microscope equipped with a ×100 objective (Plan Apo VC ×100 Oil DIC N2, Nikon). Images were acquired and analyzed with NIS Elements AR 4 software (Nikon).

## Western blot and proteomics sample preparation

HEK293/CHO cells were washed once with PBS without $Ca^{2+}$, harvested, and resuspended in RIPA lysis buffer containing (in mM) Tris (20, pH 7.4), EDTA (1), NaCl (150), 0.1% (wt/vol) SDS, 1% Triton X-100, 1% sodium deoxycholate, and supplemented with protease inhibitor mixture (10 µL/mL, Sigma-Aldrich, St. Louis, MO, USA), PMSF (1 mM, Sigma-Aldrich) and Phosstop phosphatase inhibitor cocktail tablets (Sigma-Aldrich, St. Louis, MO, USA). Lysates were prepared by incubation at 4°C for 1 hr, with occasional vortex, and cleared by centrifugation (10,000×$g$, 10 min, 4°C). Supernatants were transferred to new tubes, with aliquots removed for quantification of total protein concentration determined by the bis-cinchonic acid protein estimation kit (Pierce Technologies, Waltham, MA, USA).

For immunoprecipitation, lysates were pre-cleared by incubation with 20 µL Protein A/G Sepharose beads (Rockland) bound to anti-Rabbit IgG (Sigma) for 3 hr at 4°C. Equivalent total protein amounts were added to spin-columns containing 75 µL Protein A/G Sepharose beads incubated with 4 µg anti-Q1 (Alomone, Jerusalem, Israel), tumbling overnight at 4°C. Immunoprecipitates were washed three to five times with RIPA buffer, spun down at 500×$g$, eluted with 40 µL of warmed sample buffer (50 mM Tris, 10% [vol/vol] glycerol, 2% SDS, 100 mM DTT, and 0.2 mg/mL bromophenol blue), and boiled (55°C, 15 min). Proteins were resolved on a 4–12% Bis·Tris gradient precast gel (Life Technologies) in MOPS-SDS running buffer (Life Technologies) at 200 V constant for ~1 hr. We loaded 10 µL of the PageRuler Plus Prestained Protein Ladder (10–250 kDa, Thermo Fisher, Waltham, MA, USA) alongside the samples.

For proteomic analysis, the gels were stained with SimplyBlue (Thermo Fisher Scientific) and Q1 monomer and dimer bands were excised. In-gel digestion was performed as previously described (*Shevchenko et al., 2006*) with minor modifications. Gel slices were washed with 1:1 acetonitrile and 100 mM ammonium bicarbonate for 30 min then dehydrated with 100% acetonitrile for 10 min until shrunk. The excess acetonitrile was removed, gel slices were dried in speed-vacuum at RT for 10 min and then reduced with 5 mM DTT for 30 min at 56°C in an air thermostat, cooled down to RT, and alkylated with 11 mM IAA for 30 min with no light. Gel slices were then washed with 100 mM of ammonium bicarbonate and 100% acetonitrile for 10 min each. Excess acetonitrile was removed and dried in a speed-vacuum for 10 min at RT and the gel slices were re-hydrated in a solution of 25 ng/µL trypsin in 50 mM ammonium bicarbonate for 30 min on ice and digested overnight at 37°C in an air thermostat. Digested peptides were collected and further extracted from gel slices in extraction buffer (1:2 ratio by volume of 5% formic acid:acetonitrile) at high speed, shaking in an air thermostat. The supernatants from both extractions were combined and dried in a speed-vacuum. Peptides were dissolved in 3% acetonitrile/0.1% formic acid.

For western blotting, protein bands were transferred from the gel by tank transfer onto a nitrocellulose membrane (3.5 hr, 4°C, 30 V constant) in transfer buffer (25 mM Tris pH 8.3, 192 mM glycine, 15% [vol/vol] methanol, and 0.1% SDS). The membranes were blocked with a solution of 5% nonfat milk (Bio-Rad) in Tris-buffered saline-Tween (TBS-T) (25 mM Tris pH 7.4, 150 mM NaCl, and 0.1% Tween-20) for 1 hr at RT and then incubated overnight at 4°C with primary antibody (rabbit anti-phospho-KCNQ1) at 1:250 dilution in blocking solution. The blots were washed with TBS-T three times for 10 min each and then incubated with secondary horseradish peroxidase-conjugated antibody for 1 hr at RT. After washing in TBS-T, the blots were developed with a chemiluminescent detection kit (Pierce Technologies) and then visualized on a gel imager. Membranes were then stripped with harsh stripping buffer (2% SDS, 62 mM Tris pH 6.8, 0.8% β-mercaptoethanol) at 50°C for 30 min, rinsed under running water for 2 min, and washed with TBST (3×, 10 min). Membranes were pre-treated with 0.5% glutaraldehyde and re-blotted with rabbit anti-KCNQ1 antibody (1:1000, Alomone labs, Israel) and rabbit anti-actin antibody (1:2000, Abcam, USA).

## Liquid chromatography with tandem mass spectrometry

Desalted peptides were injected in an EASY-Spray PepMap RSLC C18 50 cm × 75 cm ID column (Thermo Scientific) connected to an Orbitrap Fusion Tribrid (Thermo Scientific). Peptides elution and separation were achieved at a non-linear flow rate of 250 nL/min using a gradient of 5–30% of buffer B (0.1% [vol/vol] formic acid, 100% acetonitrile) for 110 min with a temperature of the column maintained at 50°C during the entire experiment. The Thermo Scientific Orbitrap Fusion Tribrid mass spectrometer was used for peptide tandem mass spectroscopy (MS/MS). Survey scans of peptide precursors are performed from 350 to 1500 m/z at 120 K full width at half maximum resolution (at 200 m/z) with a $2 \times 10^5$ ion count target and a maximum injection time of 60 ms. The instrument was set to run in top speed mode with 3 s cycles for the survey and the MS/MS scans. After a survey scan, MS/MS was performed on the most abundant precursors, that is, those exhibiting a charge state from 2 to 6 of greater than $5 \times 10^3$ intensity, by isolating them in the quadrupole at 1.6 Th. We used higher-energy C-trap dissociation with 30% collision energy and detected the resulting fragments with the rapid scan rate in the ion trap. The automatic gain control target for MS/MS was set to $5 \times 10^4$ and the maximum injection time was limited to 30 ms. The dynamic exclusion was set to 30 s with a 10 ppm mass tolerance around the precursor and its isotopes. Monoisotopic precursor selection was enabled.

## LC-MS/MS data analysis

Raw mass spectrometric data were analyzed using the Proteome Discoverer 2.4 to perform database search and LFQ quantification at default settings. PD2.4 was set up to search with the reference human proteome database downloaded from UniProt and performed the search trypsin digestion with up to two missed cleavages. Peptide and protein false discovery rates were all set to 1%. The following modifications were used for protein identification and LFQ quantification: carbamidomethyl(C) was set as fixed modification and variable modifications of oxidation (M) and acetyl (protein N-term), DiGly (K), and deamination for asparagine or glutamine (NQ). Results obtained from PD2.4 were further used to quantify relative phosphorylated peptide abundances under the different conditions and identifying sites modified on KCNQ1.

## Data analysis

Patch clamp data, shown as mean ± SEM, were acquired using pCLAMP 8.0 (Axon Instruments) and analyzed with Origin 7.0 (OriginLab, Northampton, MA, USA) and Clampfit 8.2 (Axon Instruments). Flow cytometry data were analyzed using FlowJo 10.8 software. Statistical data analysis was assessed with Student's t-test for comparison between two groups and one-way ANOVA for comparisons among more than two groups, followed by pairwise comparisons using Tukey HSD post hoc test. In the figures, data are shown as mean ± SEM, and statistically significant differences to control values are indicated by symbols and described in the figure legends.

## Material availability statement

Plasmid constructs for non-commercial purposes can be obtained by request from the corresponding author after publication of the manuscript.

## Acknowledgements

We thank Dr. Ming Chen for technical support. The work was supported by NIH grants R01 HL142111 and R01 HL122421 (to HMC) and R01 GM109763 (to RSK). SAK was supported by a Medical Scientist Training Program grant (T32 GM007367) and NHLBI National Research Service Award (1F30-HL140878). Flow cytometry experiments were performed in CCTI Flow Cytometry Core, supported in part by the NIH (S10RR027050). Confocal images were collected in the HICC Confocal and Specialized Microscopy Shared Resource, supported by the NIH (P30 CA013696).

## Additional information

#### Competing interests

Henry M Colecraft: Reviewing editor, *eLife*. The other authors declare that no competing interests exist.

## Funding

| Funder | Grant reference number | Author |
|---|---|---|
| National Heart, Lung, and Blood Institute | R01 HL142111 | Henry M Colecraft |
| National Heart, Lung, and Blood Institute | R01 HL122421 | Henry M Colecraft |
| National Institutes of Health | R01 GM109763 | Robert S Kass |
| National Heart, Lung, and Blood Institute | RO1 HL121253 | Henry M Colecraft |

The funders had no role in study design, data collection and interpretation, or the decision to submit the work for publication.

## Author contributions

Xinle Zou, Sri Karthika Shanmugam, Conceptualization, Data curation, Formal analysis, Investigation, Writing - original draft, Writing - review and editing; Scott A Kanner, Conceptualization, Data curation, Formal analysis, Investigation, Writing - review and editing; Kevin J Sampson, Conceptualization, Supervision, Writing - review and editing; Robert S Kass, Conceptualization, Supervision, Funding acquisition, Writing - review and editing; Henry M Colecraft, Conceptualization, Data curation, Formal analysis, Supervision, Funding acquisition, Writing - original draft, Writing - review and editing

## Author ORCIDs

Henry M Colecraft http://orcid.org/0000-0002-2340-8899

## Decision letter and Author response

Decision letter https://doi.org/10.7554/eLife.83466.sa1
Author response https://doi.org/10.7554/eLife.83466.sa2

# Additional files

## Supplementary files

• MDAR checklist

## Data availability

All data generated or analyzed during this study are included in the manuscript, figures and associated source data files.

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
