## [Editor Report]

This important study finds that the location of recruitment of a protein kinase to an ion channel can change the complement of residues modified, leading to channel function being altered by a qualitatively distinct mechanism. Evidence for this major claim is compelling.

---

## [Decision Letter]

**Decision letter after peer review:**

Thank you for submitting your article "Divergent regulation of KCNQ1/E1 by targeted recruitment of protein kinase A to distinct sites on the channel complex" for consideration by *eLife*. Your article has been reviewed by 3 peer reviewers, including Jon Sack as Reviewing Editor and Reviewer #1, and the evaluation has been overseen by Kenton Swartz as the Senior Editor.

Essential revisions:

1) Rigor and reproducibility need to be more thoroughly addressed as noted in reviewer comments, especially concerning modest effects.

2) Please increase the rigor of testing for intracellular localization of channels or temper claims and reduce the tone in the discussion/conclusion. Discuss the relevance of mechanisms involving decreased forward trafficking to the plasma membrane versus increased internalization.

3) Please provide more controls and analysis related to the KCNE1 targeting of PKA domains, or temper claims.

*Reviewer #1 (Recommendations for the authors):*

The evidence that targeting PKA domains to the C-terminus of KCNQ1 leads to distinct phosphorylation and function seems compelling and multifaceted and could be the centerpiece of this study to an even greater extent. The controls for KCNQ1 targeting and related function are mostly outstanding. T198A and rapamycin are especially keen controls.

The context of the authors' surprise that targeted recruitment to the C-terminus of Q1 yielded a qualitatively different outcome deserves its own section of the introduction and discussion to help readers calibrate the importance of these findings. What precedent is there for a kinase creating distinct phosphorylation patterns on a single protein depending on the submolecular location of recruitment?

A major recommendation for this manuscript is to provide more controls and analysis related to the KCNE1 targeting of PKA domains, or temper claims of KCNE1 targeting. First, the effect sizes related to KCNE1 targeting seem to be within the cell-to-cell variation, and further elaboration on methods and statistical analyses could help assess the validity of interpretations. Second, controls seem to be missing that could test whether the effects of KCNE1 targeting of PKA subunits specifically result from the nanobody domains binding YFP. Specific instances are noted below.

Specific comments:

Figure 1 Expression of nano-RIIalpha and untagged Q1/E1 seems to be an important (missing) control for the claim of E1 targeting.

Figure 1 F While SEMs are clearly different, the effect is small. Providing further statistics and elaborating on details of how controls were measured for comparison (interleaved, blinded, etc.) could allay skepticism.

Page 5 "Targeting PKA-Cα to … E1 with a nanobody" Not clear that PKA-Cα is functionally targeted to E1. Suggest elaborating on evidence for this claim.

Figure 2B, C, D Clarify whether yotaio was in B, C, D, or just D.

Figure 2B, F Seeming n=1 western blots provide only weak support for phosphorylation of S27. Evidence could be bolstered by repeats and densitometry.

Figure 2D Given the overlap of individual tail decay taus, suggest using a statistic for comparison.

Page 5 "… a small augmentation of basal current amplitude (Figure 2H). These effects were similar to those observed with simple over-expression of free Cα." Basal current amplitude with simple over-expression of free Cα was not compared.

Page 5 "…a small augmentation of basal current amplitude (Figure 2H) … did not impact whole-cell current density compared to control (Figure 2 —figure supplement 1)." Unclear why Figure 2H is considered augmentation and Figure 2 —figure supplement 1 is no impact.

Figure 3 Differences between E1 {plus minus} YFP seem minor (especially compared to panel I). Suggest quantitating meanBTX-647 results with E1-YFP in a similar fashion as was done without. Clarifying # of repeats of flow cytometry runs, and run-to-run variability could help.

Figure 4 – the images from an n=1 cell under each condition are consistent with the narrative, but not compelling, and on their own provide only very weak support for the claim of ER and Golgi localization. To test the claim of colocalization with ER or Golgi markers, quantitation (e.g. colocalization statistics) and evidence of reproducibility across multiple cells/transfections would be helpful.

Page 6 "Similar results were obtained with other configurations of FKBP and FRB fusion proteins. " Clarify what similar results and other configurations are alluded to.

Figure 5D Explain more clearly how this data was normalized.

Page 8 "PHICS are likely to yield more diverse and nuanced outcomes, in part because the complement of residues modified may depend on the targeting site." Is this claim based solely on the interpretation of results here, or is there another precedent?

Page 11 "Pipette series resistance was typically 1.5 – 3 MΩ when filled with an internal solution." Please state what pipette series resistances were in whole cell mode and what range of series-resistance-induced voltage errors cells were tolerated in the analyzed data.

*Reviewer #2 (Recommendations for the authors):*

There are a few suggestions that would further strengthen the report.

1. The main concern of this study is its rigor (or the clarity of statistics throughout). In the methods section, there is a short explanation of the statistics used in this work, but it is not clear for all Figures what (and how) analyses were used. It should be clearly shown what set of data supports each claim with appropriate statistics. The authors state increase/decrease of current amplitude, I/V shifts, 'dramatic' decrease in fluorescence, etc showing only exemplar data but not summaries of data in some cases and/or summaries of data (e.g. kinetics) but not current examples (specific points described below). For instance, one major claim of the study is that targeted phosphorylation of E1, but not Q1, enhances IKs channel complex activity. However, the claim of enhanced IKs activity upon either free Cα or Cα targeted to E1-YFP approaches is very modest (at least what Figures 2C and G show). Part of this concern might be solved by showing appropriate statistical analysis throughout.

2. Along this line, is the cAMP-mediated enhancement of IKs activity at physiological voltages in the heart?

3. Another problem is the artificial nature of these systems and reproducibility (or use) in native conditions are concerning.

4. Furthermore, it is not clear if the reduced cell surface levels of IKs it is either by decreasing its forward trafficking to the plasma membrane or by increasing its internalization. This is important as the authors claim that their approach will have therapeutic relevance to rescue abnormal PKA signaling for some disease-associated AKAP mutations.

5. To me, speeding up the kinetics of IKs channel might be more physiologically relevant than increasing current amplitude (triggered by sympathetic activation of IKs in the heart during exercise or the fight-or-flight response) and Figure 1E, right does not show changes in IKs kinetics after cAMP/OA whatsoever. Moreover, in Figure 1E, right: the increase of IKs current amplitude upon cAMP/OA application is very modest: is the red trace significantly different from the black one in E-F?

6. Western blotting and electrophysiology showing Q1 phosphorylation and cAMP-mediated IKs activity upon either free Cα or Cα targeted to E1-YFP, respectively, is very modest (at least what Figure 2C and G shows). Readers would benefit from relative quantification ratios from blots shown in Figure 2B/F. Furthermore, authors should show appropriate statistical analysis supporting the claim of "enhanced IKs channel complex activity", which concomitantly will strengthen their conclusions.

7. Page 4, 3rd paragraph: exemplar IKs did not show an increase in current amplitude after breakthrough to the whole-cell configuration if the pipette solution lacked cAMP/OA (Figure 1E, left). In sharp contrast, when cAMP/OA was present in the patch pipette, exemplar IKs displayed a significant increase in current after three minutes of dialysis with intracellular solution (Figure 1E, right). Where do authors show statistical analysis backing up this claim? The only place showing statistics in this figure is in panel (i).

8. On Page 5, 2nd paragraph: "…we compared key gaiting parameters of currents…". I agree gating parameters (e.g. kinetics, see point 1) should be thoroughly analyzed. While Figure 2D showed Taudeact for tail currents, there are no original current traces supporting this summarized data. This should be shown. Likewise, original traces supporting data in panels E-H should be added.

9. Along this line, on Page 5, 2nd paragraph: "Compared to control cells expressing Q1 + E1-YFP + nano, channels co-expressed with nanoCα displayed an increased phosphorylation of Q1 (Figure 2F)" no statistics backing this (see point above). And next "…, a leftward shift in the voltage-dependence of activation (V0.5,act = 34.1 {plus minus} 4.4 mV, n = 10 for Q1 + E1-YFP + nano and V0.5,act = 25.2 {plus minus} 3.4, n = 10 for Q1 + E1-YFP + nano Cα, p = 0.049) (Figure 2G), and a small augmentation of basal current.

10. Amplitude (Figure 2H)." but Figure legend does not show statistics and Figure 2H does not show what test was used.

11. Page 6, 1st paragraph: "…a dramatic decrease in BTX-647 fluorescence compared to nano control…". I don't see a reason for this "dramatic" change when no statistical analysis where done/shown in Figure 3H-I.

12. Page 6. 1st paragraph at the end: "(Figure 3, H and I), indicating that recruiting these PKA subunits to Q1 C-terminus suppresses channel surface trafficking, and rationalizing the inhibitory impact on IKs".

13. Why not the enhancement of channel retrieval from the membrane?

14. The authors should explain better the section "Inducible recruitment of nanoCα to Q1 reveals slow temporal regulation of channel trafficking" on page 6. To me, the narrative of this part of the paper is insufficiently and/or convolutely described and it is hard to follow. Please define what is FKBP and FRB etc.

15. Overall, it is not clear if the reduced cell surface levels of IKs it is either by decreasing its forward trafficking to the plasma membrane or by increasing its internalization.

16. Would these findings be mirrored in cardiomyocytes? It has been shown that the heart, besides expressing KCNQ1 and KCNE1 subunits, also expresses the KCNQ1-modulatory auxiliary subunits KCNE2 and KCNE3 (PMID: 15698834). Functional consequences of IKs activity upon phosphorylation of residue S27 in KCNQ1 vary depending on the presence of KCNE1 and KCNE3 in the channel complexes. Furthermore, on its own, KCNE1 modulates KCNQ1 internalization through clathrin-mediated endocytosis of KCNQ1 (PMID: 19202166). How these facts affect the conclusion should be discussed.

---

## [Author Response]

Essential revisions:1) Rigor and reproducibility need to be more thoroughly addressed as noted in reviewer comments, especially concerning modest effects.

We have responded to the reviewer comments on rigor and reproducibility by performing additional experiments and analyses. The changes made are detailed in the response to reviewer critiques below.

2) Please increase the rigor of testing for intracellular localization of channels or temper claims and reduce the tone in the discussion/conclusion. Discuss the relevance of mechanisms involving decreased forward trafficking to the plasma membrane versus increased internalization.

We have performed Pearson’s correlation analyses to address the concern regarding the rigor of the confocal evidence regarding the retention of channels in intracellular compartments under different conditions. The details are provided in the response to reviewer critiques below.

3) Please provide more controls and analysis related to the KCNE1 targeting of PKA domains, or temper claims.

We have performed additional experiments and analyses to bolster the claims regarding the functional effects of targeting PKA subunits to KCNE1. The details of the changes made are provided in the response to reviewer critiques.

Reviewer #1 (Recommendations for the authors):Specific comments:Figure 1 Expression of nano-RIIalpha and untagged Q1/E1 seems to be an important (missing) control for the claim of E1 targeting.

We thank the reviewer for this helpful feedback. We have now included this control condition in Figure 1 —figure supplement 1. The data shows that there is no cAMP-mediated upregulation of *I*_Ks_ current density by nano-RII*α* when KCNE1 is not tagged with YFP, highlighting the need for E1 targeting to achieve localized PKA response on the channel.

Figure 1 F While SEMs are clearly different, the effect is small. Providing further statistics and elaborating on details of how controls were measured for comparison (interleaved, blinded, etc.) could allay skepticism.

We have added statistical analyses to support the claim of a significant difference to the data in Figure 1F. The relevant text in the revised manuscript (p4) reads:

“In population time course data, the temporal evolution of normalized current amplitude displayed a clear-cut divergence between recordings obtained with or without cAMP/OA in the patch pipette solution (*I*_3min_/*I*_0_ = 0.9615 ± 0.025, *n* = 14 for Q1 + E1-YFP + nanoRII*α* without cAMP + OA; *I*_3min_/*I*_0_ = 1.0812 ± 0.030, n = 12, for Q1 + E1-YFP + nanoRII*α* with cAMP + OA; *P* = 0.0054, unpaired t-test) (Figure 1G). Controls were measured interleaved with the experiment group.”

Moreover, the magnitude of the difference in normalized *I*_Ks_ amplitude shown in Figure 1F ±cAMP/OA in cells expressing Q1/E1 + nanoRIIa is similar to what has been previously published for Q1/E1 ± AKAP-9 (e.g. Kurokawa et al., 2004, PNAS). We have now noted this in the revised manuscript (p4):

“The magnitude of the observed response in the diary plots is comparable to the normalized enhancement of IKs current observed with cAMP + OA in cells expressing Q1 + E1 + AKAP9 ^30^.”

Page 5 "Targeting PKA-Cα to … E1 with a nanobody" Not clear that PKA-Cα is functionally targeted to E1. Suggest elaborating on evidence for this claim.

To further substantiate the claim of functional targeting of PKA-C*α* to E1 by means of a YFP nanobody (nanoC*α)*, we did a confirmatory immunoprecipitation test. We co-expressed Q1 and E1-YFP with either free Cα or nanoCα, performed a pulldown with anti-Cα, and probed for E1 using anti-YFP. We found that nanoCα robustly pulled down E1-YFP while free PKA-C*α* comparatively did not, explicitly demonstrating the targeting efficiency of nanoCα to E1-YFP in the channel complex. This data is now shown in Figure 2 —figure supplement 1 and described I the revised text (p5):

“In this configuration, Cα is recruited to the tagged E1 subunit in the channel complex (Figure 2E), as confirmed in a pull-down assay (Figure 2 —figure supplement 1).”

Figure 2B, C, D Clarify whether yotaio was in B, C, D, or just D.

Yotiao was present in just D.

Figure 2B, F Seeming n=1 western blots provide only weak support for phosphorylation of S27. Evidence could be bolstered by repeats and densitometry.

We agree with the reviewer and have quantified the increase in phosphorylated Q1 in response to free Cα and nanoCα by computing the normalized fold-change of pQ1/Q1 in cells expressing Q1 + E1-YFP + Cα (or nanoCα) relative to those expressing Q1 + E1-YFP + nano (or pcDNA3). This is described in the text (p5) as:

“Indeed, Western blotting indicated that co-expressing Cα led to an increase in Q1 phosphorylation (normalized pQ1/Q1 is increased 1.516-fold for cell expressing Cα compared to control)…”

And,

“Compared to control cells expressing Q1 + E1-YFP + nano, channels co-expressed with nanoCα displayed an increased phosphorylation of Q1 (normalized pQ1/Q1 is increased 2.2-fold in cells co-expressing nanoCα compared to controls expressing nano) (Figure 2F)….”

Unfortunately, the pQ1 antibody was obtained by the Kass lab and used for various studies over several years and we were unable to procure more of the antibody for more repeats of the experiments. Obtaining more of the antibody by animal injection and rigorously testing it is a lengthy process that would be beyond the scope of the current revisions.

Figure 2D Given the overlap of individual tail decay taus, suggest using a statistic for comparison.

We thank the reviewer for the suggestion. One-way ANOVA analysis indicated that the differences in tail decay taus just fell short of statistical significance (*P* = 0.053216, One-way ANOVA). Accordingly, we have revised the text (p5) which reads:

“….and a trend towards a slower rate of tail current deactivation (in the presence of yotiao) (Figure 2D), two signatures of PKA regulation of *I*_Ks_ that is mediated via phosphorylation of Ser27 in Q1. Consistent with this, Q1[S27A] reversed the trend towards Cα-induced decreased rate of tail current deactivation observed with wild-type Q1 (Figure 2D).”

Page 5 "… a small augmentation of basal current amplitude (Figure 2H). These effects were similar to those observed with simple over-expression of free Cα." Basal current amplitude with simple over-expression of free Cα was not compared.

We thank the reviewer for catching this. We have removed the reference to the impact of free Cα on basal current amplitude in the revised manuscript.

Page 5 "…a small augmentation of basal current amplitude (Figure 2H) … did not impact whole-cell current density compared to control (Figure 2 —figure supplement 1)." Unclear why Figure 2H is considered augmentation and Figure 2 —figure supplement 1 is no impact.

We thank the reviewer for pointing out this potential source of confusion. In Figure 2 —figure supplement 2 we show that recruiting nanoCα to TASK-GFP did not decrease whole-cell current in contrast to our observations when we recruited nanoCα to KCNQ1-YFP (Figure 2, J and K). We have revised the text to clarify this point (p5):

“Co-expressing nanoCα with TASK-4-GFP, a two-pore domain K^+^ channel, did not decrease whole-cell current density compared to control (Figure 2 —figure supplement 2).”

Figure 3 Differences between E1 {plus minus} YFP seem minor (especially compared to panel I). Suggest quantitating meanBTX-647 results with E1-YFP in a similar fashion as was done without. Clarifying # of repeats of flow cytometry runs, and run-to-run variability could help.

We thank the reviewer for these helpful suggestions. We have now included the total number of repeats and meanBTX647 values for the flow cytometry experiments in Figure 3. The relevant new sentences in the text (p6) now read:

“Co-expressing nanoCα resulted in an elevated BTX-647 fluorescence signal (mean_BTX-647_ = 1,592 ± 41.6 a.u., *n* = 19,331, *N* = 2 for BBS-Q1 + E1 + nano; and mean_BTX-647_ = 2,513 ± 56.0 a.u., *n* = 21,822, *N* = 2 for BBS-Q1 + E1 + nanoCα; *P* < 0.001, one-way ANOVA and Tukey HSD post-hoc test) indicating an increase in channel surface density, while nanoRIIα had a more modest impact (mean_BTX-647_ = 1,905 ± 43.19 a.u., *n* = 18,343, *N* = 2 for BBS-Q1 + E1 + nanoRIIα) (Figure 3A-C). Next, we examined how tethering of nano, nanoCα, or nanoRIIα to E1-YFP affected channel trafficking. In this configuration, both nanoCα and nanoRIIα resulted in an increase in BTX-647 fluorescence compared to nano control (mean_BTX-647_ = 1557 ± 48.04 a.u., *n* = 18,543, *N* = 2 for BBS-Q1 + E1-YFP + nano; and mean_BTX-647_ = 1,969 ± 90.50 a.u., *n* = 20,843, *N* = 2 for BBS-Q1 + E1-YFP + nanoCα; mean_BTX-647_ = 2,290 ± 47.74 a.u., *n* = 16,863, *N* = 2 for BBS-Q1 + E1-YFP + nanoRIIα, *P* < 0.001, one-way ANOVA and Tukey HSD post-hoc test) (Figure 3, D-F).”

And,

“Both nanoCα and nanoRIIα led to a dramatic decrease in BTX-647 fluorescence compared to nano control (mean_BTX-647_ = 2,672 ± 74.3 a.u., *n* = 14,405, *N* = 4 for BBS-Q1-YFP + E1 + nano; mean_BTX-647_ = 406 ± 51.1 a.u., *n* = 16,514, *N* = 4 for BBS-Q1-YFP + E1 + nanoCα; and mean_BTX-647_ = 543 ± 25.6 a.u., *n* = 13,870, *N* = 4 for BBS-Q1-YFP + E1 + nanoRIIα, *P <* 0.001, one-way ANOVA and Tukey HSD post-hoc test) (Figure 3, H and I), indicating that recruiting these PKA subunits to Q1 C-terminus suppresses channel surface density, and rationalizing the inhibitory impact on *I*_Ks_.”

Figure 4 – the images from an n=1 cell under each condition are consistent with the narrative, but not compelling, and on their own provide only very weak support for the claim of ER and Golgi localization. To test the claim of colocalization with ER or Golgi markers, quantitation (e.g. colocalization statistics) and evidence of reproducibility across multiple cells/transfections would be helpful.

We have now modified Figure 4 to include Pearson’s colocalization coefficient analysis to quantify KCNQ1-YFP colocalization with ER-mCherry and Golgi-mCherry markers. The added analysis is described in the Results section (p6,7):

“By contrast, co-expressing either nanoCα or nanoRIIα significantly increased the co-localization of Q1-YFP with ER-mCherry (Pearson’s colocalization coefficient (PCC) = 0.85 ± 0.020, *n* = 9 for nano; PCC = 0.90 ± 0.015, *n* = 7 for nanoCα; and PCC = 0.94 ± 0.010, *n* = 8 for nanoRIIα; *P* = 0.0049, one-way ANOVA and Tukey HSD post-hoc test) (Figure 4, A and B) and Golgi-mCherry (PCC = 0.84 ± 0.016, n = 9 for nano; PCC = 0.90 ± 0.012, *n* = 8 for nanoCα; and PCC = 0.90 ± 0.011, *n* = 6 for nanoRIIα; *P* = 0.0065, one-way ANOVA and Tukey HSD post-hoc test) (Figure 4, C and D), respectively, with no discernible YFP signal at the cell surface. Thus, targeting Cα or RIIα to Q1-YFP leads to increased retention of the channel in the ER and Golgi compartments.”

Page 6 "Similar results were obtained with other configurations of FKBP and FRB fusion proteins. " Clarify what similar results and other configurations are alluded to.

We have removed the sentence regarding other configurations of FKBP and FRB fusion proteins.

Figure 5D Explain more clearly how this data was normalized.

For Figure 5D, the mean YFP fluorescence at the different time points was computed and divided by the mean YFP fluorescence at t=0 to measure changes in the normalized in Q1-YFP expression over time.

Page 8 "PHICS are likely to yield more diverse and nuanced outcomes, in part because the complement of residues modified may depend on the targeting site." Is this claim based solely on the interpretation of results here, or is there another precedent?

We are unaware of any previous precedent where the same kinase recruited to distinct sites on a macromolecular complex yielded divergent functional outcomes. Thus, the sentence referred to is based on interpretation of the results presented here.

Reviewer #2 (Recommendations for the authors):There are a few suggestions that would further strengthen the report.1. The main concern of this study is its rigor (or the clarity of statistics throughout). In the methods section, there is a short explanation of the statistics used in this work, but it is not clear for all Figures what (and how) analyses were used. It should be clearly shown what set of data supports each claim with appropriate statistics. The authors state increase/decrease of current amplitude, I/V shifts, 'dramatic' decrease in fluorescence, etc showing only exemplar data but not summaries of data in some cases and/or summaries of data (e.g. kinetics) but not current examples (specific points described below). For instance, one major claim of the study is that targeted phosphorylation of E1, but not Q1, enhances IKs channel complex activity. However, the claim of enhanced IKs activity upon either free Cα or Cα targeted to E1-YFP approaches is very modest (at least what Figures 2C and G show). Part of this concern might be solved by showing appropriate statistical analysis throughout.

We thank the reviewer for their valuable feedback, we have now included statistical information for all the figures individually as suggested to improve clarity. Changes made in the text are indicated for specific questions below.

2. Along this line, is the cAMP-mediated enhancement of IKs activity at physiological voltages in the heart?

The cAMP-mediated enhancement of IKs serves to limit the increase in action potential duration that would otherwise occur due to increased L-type calcium current. So, yes, this regulation occurs at physiological voltages in the heart.

3. Another problem is the artificial nature of these systems and reproducibility (or use) in native conditions are concerning.

This study was conceived to design and test a novel approach to reconstitute I_Ks_ modulation using PHICS. While it is true that the experiments were done in heterologous systems, the findings serve as proof-of-concept to potentially develop these tools as therapeutics. We have added a sentence noting this limitation and the importance of future studies in native conditions (p8/9):

“This approach would require the generation of a nanobody or other antibody-mimetic that binds to the intracellular domain of E1.”

4. Furthermore, it is not clear if the reduced cell surface levels of IKs it is either by decreasing its forward trafficking to the plasma membrane or by increasing its internalization. This is important as the authors claim that their approach will have therapeutic relevance to rescue abnormal PKA signaling for some disease-associated AKAP mutations.

Because nanoCα and nanoRIIα substantially decrease the steady-state surface density of Q1-YFP, it is challenging to experimentally determine whether the effect is due to decreased forward trafficking or increased endocytosis (or both) due to unfavorable signal to noise. We speculate that the reduced surface density by targeting C*α* to KCNQ1 results in decreased forward trafficking due to the increase in its colocalization with ER and Golgi markers (Figure 4) and the length of time it takes for the decrease in surface density to become evident using the rapamycin-mediated induced recruitment approach (Figure 5).

5. To me, speeding up the kinetics of IKs channel might be more physiologically relevant than increasing current amplitude (triggered by sympathetic activation of IKs in the heart during exercise or the fight-or-flight response) and Figure 1E, right does not show changes in IKs kinetics after cAMP/OA whatsoever. Moreover, in Figure 1E, right: the increase of IKs current amplitude upon cAMP/OA application is very modest: is the red trace significantly different from the black one in E-F?

We have added statistical analyses to support the claim of a significant difference to the data in Figure 1F. The relevant text in the revised manuscript (p4) reads:

“In population time course data, the temporal evolution of normalized current amplitude displayed a clear-cut divergence between recordings obtained with or without cAMP/OA in the patch pipette solution (*I*_3min_/*I*_0_ = 0.9615 ± 0.025, *n* = 14 for Q1 + E1-YFP + nanoRII*α* without cAMP + OA; *I*_3min_/*I*_0_ = 1.0812 ± 0.030, n = 12, for Q1 + E1-YFP + nanoRII*α* with cAMP + OA; *P* = 0.0054, unpaired t-test) (Figure 1G). Controls were measured interleaved with the experiment group.”

Moreover, the magnitude of the difference in normalized *I*_Ks_ amplitude shown in Figure 1F ±cAMP/OA in cells expressing Q1/E1 + nanoRIIa is similar to what has been previously published for Q1/E1 ± AKAP-9 (e.g. Kurokawa et al., 2004, PNAS). We have now noted this in the revised manuscript (p4):

“The magnitude of the observed response in the diary plots is comparable to the normalized enhancement of IKs current observed with cAMP + OA in cells expressing Q1 + E1 + AKAP9 ^30^.”

6. Western blotting and electrophysiology showing Q1 phosphorylation and cAMP-mediated IKs activity upon either free Cα or Cα targeted to E1-YFP, respectively, is very modest (at least what Figure 2C and G shows). Readers would benefit from relative quantification ratios from blots shown in Figure 2B/F. Furthermore, authors should show appropriate statistical analysis supporting the claim of "enhanced IKs channel complex activity", which concomitantly will strengthen their conclusions.

We agree with the reviewer and have quantified the increase in phosphorylated Q1 in response to free Cα and nanoCα by computing the normalized fold-change of pQ1/Q1 in cells expressing Q1 + E1-YFP + Cα (or nanoCα) relative to those expressing Q1 + E1-YFP + nano (or pcDNA3). This is described in the text (p5) as:

“Indeed, Western blotting indicated that co-expressing Cα led to an increase in Q1 phosphorylation (normalized pQ1/Q1 is increased 1.516-fold for cell expressing Cα compared to control)…”

And,

“Compared to control cells expressing Q1 + E1-YFP + nano, channels co-expressed with nanoCα displayed an increased phosphorylation of Q1 (normalized pQ1/Q1 is increased 2.2-fold in cells co-expressing nanoCα compared to controls expressing nano) (Figure 2F)….”

Unfortunately, the pQ1 antibody was obtained by the Kass lab and used for various studies over several years and we were unable to procure more of the antibody for more repeats of the experiments. Obtaining more of the antibody by animal injection and rigorously testing it is a lengthy process that would be beyond the scope of the current revisions.

We have included statistical analyses for the figures in Figure 2 which are described in the Results (p5) as:

“Current vs voltage (*I*-*V*) curves indicated that by comparison to currents obtained with Q1 + E1-YFP alone, those recorded from cells co-expressing Cα displayed a hyperpolarizing shift in the voltage-dependence activation (*V*_0.5,act_ = 34.5 ± 3.6 mV, *n* = 13 for Q1 + E1-YFP and *V*_0.5,act_ = 25.0 ± 2.7, *n* = 13 for Q1 + E1-YFP + Cα, *P* = 0.02, unpaired *t* test) (Figure 2C), and a trend towards a slower rate of tail current deactivation (in the presence of yotiao) (Figure 2D)….”

And,

“Compared to control cells expressing Q1 + E1-YFP + nano, channels co-expressed with nanoCα displayed an increased phosphorylation of Q1 (normalized pQ1/Q1 is increased 2.2-fold in cells co-expressing nanoCα compared to controls expressing nano) (Figure 2F), a leftward shift in the voltage-dependence of activation (*V*_0.5,act_ = 34.1 ± 4.4 mV, *n* = 10 for Q1 + E1-YFP + nano and *V*_0.5,act_ = 25.2 ± 3.4, *n* = 10 for Q1 + E1-YFP + nanoCα, *p* = 0.049, unpaired, *t* test) (Figure 2G), and a trend towards a small augmentation of basal current amplitude (*I*_avg_ = 232.1 ± 36.44 mV, *n* = 11 for Q1 + E1-YFP + nano and *I*_avg_ = 267.24 ± 37.12, *n* = 16 for Q1 + E1-YFP + nanoCα, *P* = 0.5054)….”

7. Page 4, 3rd paragraph: exemplar IKs did not show an increase in current amplitude after breakthrough to the whole-cell configuration if the pipette solution lacked cAMP/OA (Figure 1E, left). In sharp contrast, when cAMP/OA was present in the patch pipette, exemplar IKs displayed a significant increase in current after three minutes of dialysis with intracellular solution (Figure 1E, right). Where do authors show statistical analysis backing up this claim? The only place showing statistics in this figure is in panel (i).

We have added statistical analyses to support the claim of a significant difference to the data in Figure 1F. The relevant text in the revised manuscript (p4) reads:

“In population time course data, the temporal evolution of normalized current amplitude displayed a clear-cut divergence between recordings obtained with or without cAMP/OA in the patch pipette solution (*I*_3min_/*I*_0_ = 0.9615 ± 0.025, *n* = 14 for Q1 + E1-YFP + nanoRII*α* without cAMP + OA; *I*_3min_/*I*_0_ = 1.0812 ± 0.030, n = 12, for Q1 + E1-YFP + nanoRII*α* with cAMP + OA; *P* = 0.0054, unpaired t-test) (Figure 1G). Controls were measured interleaved with the experiment group.”

Moreover, the magnitude of the difference in normalized *I*_Ks_ amplitude shown in Figure 1F ±cAMP/OA in cells expressing Q1/E1 + nanoRIIa is similar to what has been previously published for Q1/E1 ± AKAP-9 (e.g. Kurokawa et al., 2004, PNAS). We have now noted this in the revised manuscript (p4):

“The magnitude of the observed response in the diary plots is comparable to the normalized enhancement of IKs current observed with cAMP + OA in cells expressing Q1 + E1 + AKAP9 ^30^.”

8. On Page 5, 2nd paragraph: "…we compared key gaiting parameters of currents…". I agree gating parameters (e.g. kinetics, see point 1) should be thoroughly analyzed. While Figure 2D showed Taudeact for tail currents, there are no original current traces supporting this summarized data. This should be shown. Likewise, original traces supporting data in panels E-H should be added.9. Along this line, on Page 5, 2nd paragraph: "Compared to control cells expressing Q1 + E1-YFP + nano, channels co-expressed with nanoCα displayed an increased phosphorylation of Q1 (Figure 2F)" no statistics backing this (see point above). And next "…, a leftward shift in the voltage-dependence of activation (V0.5,act = 34.1 {plus minus} 4.4 mV, n = 10 for Q1 + E1-YFP + nano and V0.5,act = 25.2 {plus minus} 3.4, n = 10 for Q1 + E1-YFP + nano Cα, p = 0.049) (Figure 2G), and a small augmentation of basal current.

Please see response to point # 2 above.

10. Amplitude (Figure 2H)." but Figure legend does not show statistics and Figure 2H does not show what test was used.

We have added the statistical analysis and modified the text which now reads:

“….and a trend towards a small augmentation of basal current amplitude (*I*_avg_ = 232.1 ± 36.44 mV, *n* = 11 for Q1 + E1-YFP + nano and *I*_avg_ = 267.24 ± 37.12, *n* = 16 for Q1 + E1-YFP + nanoCα, *P* = 0.5054) (Figure 2H).”

11. Page 6, 1st paragraph: "…a dramatic decrease in BTX-647 fluorescence compared to nano control…". I don't see a reason for this "dramatic" change when no statistical analysis where done/shown in Figure 3H-I.

We thank the reviewer for these helpful suggestions. We have now included the total number of repeats and meanBTX647 values for the flow cytometry experiments in Figure 3. The relevant new sentences in the text (p6) now read:

“Co-expressing nanoCα resulted in an elevated BTX-647 fluorescence signal (mean_BTX-647_ = 1,592 ± 41.6 a.u., *n* = 19,331, *N* = 2 for BBS-Q1 + E1 + nano; and mean_BTX-647_ = 2,513 ± 56.0 a.u., *n* = 21,822, *N* = 2 for BBS-Q1 + E1 + nanoCα; *P* < 0.001, one-way ANOVA and Tukey HSD post-hoc test) indicating an increase in channel surface density, while nanoRIIα had a more modest impact (mean_BTX-647_ = 1,905 ± 43.19 a.u., *n* = 18,343, *N* = 2 for BBS-Q1 + E1 + nanoRIIα) (Figure 3A-C). Next, we examined how tethering of nano, nanoCα, or nanoRIIα to E1-YFP affected channel trafficking. In this configuration, both nanoCα and nanoRIIα resulted in an increase in BTX-647 fluorescence compared to nano control (mean_BTX-647_ = 1557 ± 48.04 a.u., *n* = 18,543, *N* = 2 for BBS-Q1 + E1-YFP + nano; and mean_BTX-647_ = 1,969 ± 90.50 a.u., *n* = 20,843, *N* = 2 for BBS-Q1 + E1-YFP + nanoCα; mean_BTX-647_ = 2,290 ± 47.74 a.u., *n* = 16,863, *N* = 2 for BBS-Q1 + E1-YFP + nanoRIIα, *P* < 0.001, one-way ANOVA and Tukey HSD post-hoc test) (Figure 3, D-F).”

And,

“Both nanoCα and nanoRIIα led to a dramatic decrease in BTX-647 fluorescence compared to nano control (mean_BTX-647_ = 2,672 ± 74.3 a.u., *n* = 14,405, *N* = 4 for BBS-Q1-YFP + E1 + nano; mean_BTX-647_ = 406 ± 51.1 a.u., *n* = 16,514, *N* = 4 for BBS-Q1-YFP + E1 + nanoCα; and mean_BTX-647_ = 543 ± 25.6 a.u., *n* = 13,870, *N* = 4 for BBS-Q1-YFP + E1 + nanoRIIα, *P <* 0.001, one-way ANOVA and Tukey HSD post-hoc test) (Figure 3, H and I), indicating that recruiting these PKA subunits to Q1 C-terminus suppresses channel surface density, and rationalizing the inhibitory impact on *I*_Ks_.”

12. Page 6. 1st paragraph at the end: "(Figure 3, H and I), indicating that recruiting these PKA subunits to Q1 C-terminus suppresses channel surface trafficking, and rationalizing the inhibitory impact on IKs".13. Why not the enhancement of channel retrieval from the membrane?

We agree with the reviewers point that we have not defined the mechanism for the decreased channel surface density. We have modified the text to state:

“(Figure 3, H and I), indicating that recruiting these PKA subunits to Q1 C-terminus suppresses channel surface density, and rationalizing the inhibitory impact on *I*_Ks_.”

14. The authors should explain better the section "Inducible recruitment of nanoCα to Q1 reveals slow temporal regulation of channel trafficking" on page 6. To me, the narrative of this part of the paper is insufficiently and/or convolutely described and it is hard to follow. Please define what is FKBP and FRB etc.

We have added a clarification to the heterodimerization strategy and defined FKBP and FRB (p6):

“To gain insights into this question we exploited a small molecule-induced heterodimerization strategy, in which rapamycin simultaneously binds to the FK506 binding protein (FKBP) and the FKBP–rapamycin binding domain (FRB) of the mammalian target of rapamycin (mTOR)”

15. Overall, it is not clear if the reduced cell surface levels of IKs it is either by decreasing its forward trafficking to the plasma membrane or by increasing its internalization.

Please see response to point #8 above.

16. Would these findings be mirrored in cardiomyocytes? It has been shown that the heart, besides expressing KCNQ1 and KCNE1 subunits, also expresses the KCNQ1-modulatory auxiliary subunits KCNE2 and KCNE3 (PMID: 15698834). Functional consequences of IKs activity upon phosphorylation of residue S27 in KCNQ1 vary depending on the presence of KCNE1 and KCNE3 in the channel complexes. Furthermore, on its own, KCNE1 modulates KCNQ1 internalization through clathrin-mediated endocytosis of KCNQ1 (PMID: 19202166). How these facts affect the conclusion should be discussed.

These are interesting and relevant questions posed by the reviewer. We hope to address and investigate these aspects in follow up studies in cardiac myocytes. In the Discussion (p9), we describe previous studies relating to phosphorylation-mediated regulation of Q1/E1 channel trafficking.